# Asymmetric reconstruction of mammalian reovirus reveals interactions among RNA, transcriptional factor μ2 and capsid proteins

Muchen Pan [1,2,3,4,5], Ana L. Alvarez-Cabrera [2,3], Joon S. Kang [2,3,6], Lihua Wang[1,7], Chunhai Fan [5] & Z. Hong Zhou [2,3,6 ✉]

Mammalian reovirus (MRV) is the prototypical member of genus *Orthoreovirus* of family *Reoviridae*. However, lacking high-resolution structures of its RNA polymerase cofactor μ2 and infectious particle, limits understanding of molecular interactions among proteins and RNA, and their contributions to virion assembly and RNA transcription. Here, we report the 3.3 Å-resolution asymmetric reconstruction of transcribing MRV and in situ atomic models of its capsid proteins, the asymmetrically attached RNA-dependent RNA polymerase (RdRp) λ3, and RdRp-bound nucleoside triphosphatase μ2 with a unique RNA-binding domain. We reveal molecular interactions among virion proteins and genomic and messenger RNA. Polymerase complexes in three *Spinoreovirinae* subfamily members are organized with different pseudo-$D_{3d}$ symmetries to engage their highly diversified genomes. The above interactions and those between symmetry-mismatched receptor-binding σ1 trimers and RNA-capping λ2 pentamers balance competing needs of capsid assembly, external protein removal, and allosteric triggering of endogenous RNA transcription, before, during and after infection, respectively.

[1] CAS Key Laboratory of Interfacial Physics and Technology, Shanghai Institute of Applied Physics, Chinese Academy of Sciences, Shanghai, China. [2] Department of Microbiology, Immunology and Molecular Genetics, University of California, Los Angeles (UCLA), Los Angeles, CA, USA. [3] California NanoSystems Institute, UCLA, Los Angeles, CA, USA. [4] University of Chinese Academy of Sciences, Beijing, China. [5] School of Chemistry and Chemical Engineering, Frontiers Science Center for Transformative Molecules, Institute of Translational Medicine, Shanghai Jiao Tong University, Shanghai, China. [6] Molecular Biology Institute, UCLA, Los Angeles, CA, USA. [7] The Interdisciplinary Research Center, Shanghai Synchrotron Radiation Facility, Zhangjiang Laboratory, Shanghai Advanced Research Institute, Chinese Academy of Sciences, Shanghai, China. ✉email: Hong.Zhou@UCLA.edu

Mammalian reovirus (MRV) infects a broad range of mammals, including humans, and affects gastrointestinal and respiratory tracts[1,2]. It also has beneficial applications, as reflected by ongoing clinical trials with engineered MRV type 3 Dearing strain in oncolytic virotherapy[3,4]. Significantly, MRV is the prototypical member of the Orthoreovirinae genus of the Reoviridae[5], a large family of double-stranded RNA (dsRNA) viruses, including the life-threatening human pathogen rotavirus[6]. These viruses differ in two main aspects: the number of protein layers of their capsids and the presence (in the viruses of nine genera that comprise the Spinoreovirinae subfamily) or absence (in the viruses of six genera that comprise the Sedoreovirinae subfamily) of mRNA-capping turrets on the twelve icosahedral vertices of their innermost capsid layer[5]. All viruses in the Reoviridae are capable of transcribing RNA inside intact capsids (endogenous transcription) without the need of host factors, an attracting characteristic that allows viral RNA transcription of these viruses to be studied in vitro in cell-free environment[2].

Among members of the Spinoreovirinae subfamily, MRV contrasts cytoplasmic polyhedrosis virus (CPV) by the number of their capsid layers: the former has two concentric protein layers[7], the latter a single shelled capsid[8,9] structurally equivalent to the MRV core[2]. The outer layer of MRV comprises protection protein σ3 and penetration protein μ1 (see ref. [2]). Removing σ3 leads to the maximally infectious form of MRV, known as the infectious subvirion particle (ISVP)[10], which uses its newly exposed μ1 penetration proteins to escape from the endosome into host cytoplasm[11]. The ISVP transcribes its ten genomic segments[12], adds a methylated guanosine cap to the 5′ end of each mRNA transcript[13], and releases the capped mRNA from the turrets at the capsid vertices[14].

The atomic models of most individual proteins[14–16] and the core[13] of MRV have been solved already by X-ray crystallography and provided insight into viral RNA transcription, capsid assembly, and host attachment not only for MRV in particular but also for dsRNA viruses in general. Historically, MRV is one of the first viruses subjected to 3D reconstruction by cryoEM[17]. In 2003, cryoEM reconstruction of MRV already reached 7.6 Å resolution and resolved secondary structures of RdRp λ3 protein[18]. Despite imposition of icosahedral symmetry in this reconstruction, fitting of existing crystal models into this reconstruction localized inside the capsid attachment sites and revealed the possible non-icosahedrally related locations of RdRp λ3 (see ref. [18]). RdRp λ3 and its protein co-factor μ2 are collectively known as the transcribing enzyme complex (TEC). Nonetheless, because imposition of icosahedral symmetry in this cryoEM reconstruction and the crystal structure of the MRV core[13] smeared asymmetrically organized elements, which of the 60 subunits of the capsid shell protein (CSP) λ1 and how the genomic RNA associate with TEC remain unknown. Biochemical and virology studies have shown that μ2 displays functionally and structurally unique sequence domains involved in discrete steps of inclusion/viroplasm development and viral replication[19,20]. RdRp's atomic structure has been solved by X-ray crystallography[16], but not μ2's, preventing a full description of the transcription mechanism, and how different domains of μ2 contribute to viral replication. MRV's receptor-binding protein σ1 forms a trimeric filament whose structure has been determined by X-ray crystallography[15], but how it is anchored to the pentameric turret remains unknown due to a lack of high-resolution in situ structures. Therefore, notwithstanding the significance of prior cryoEM and X-ray structures, the above three sets of questions, and how interactions among capsid proteins outside trigger RNA transcription inside the virus, remain unanswered in the absence of an atomic resolution description of the infectious MRV particle.

In this study, we have determined near-atomic resolution asymmetric structures of the MRV ISVP by cryo electron microscopy (cryoEM) with a sub-particle reconstruction workflow. Our results not only unveil the atomic structure of μ2 (which we show is an NTPase) but also reveal interactions among RdRp λ3, λ1, μ2, and genomic and newly transcribed RNA, as well as among external capsid proteins, including the symmetry-mismatched receptor-binding protein σ1 trimer and capping enzyme λ2 pentamer. As the high-resolution structures of MRV and of the Orthoreovirinae genus of the Reoviridae, these structures fill in a critical knowledge gap in the ever-growing repertoire of dsRNA virus structures; several structural features provide mechanistic insights into allosteric triggering and catalytic regulation of endogenous RNA transcription inside multilayered members of the Spinoreovirinae subfamily of the Reoviridae.

## Results

**Overall protein interactions and RNA genome organization.** Isolated with minimal steps and without density gradient, our MRV particles (Supplementary Fig. 1) contained a mixture of both virion and ISVP and were separated by computational classification (Supplementary Fig. 2). Due to symmetry mismatch among RNA, internal and external protein components, we implemented a step-wise sorting and symmetry-guided sub-particle reconstruction workflow (Supplementary Fig. 3) to improve the resolution of the cryoEM reconstructions of local regions (i.e., sub-particles) of the ISVP to 3.3 Å. By applying the asymmetric orientation parameters of the classified sub-particles to their corresponding full particles (see "Methods" section), we eventually reconstructed an asymmetric structure of the full ISVP at 4.3 Å resolution, revealing the global architecture and genomic RNA organization of the full infectious particle (Fig. 1a, b). Virions, which are non-infectious (quiescent) due to presence of protection protein σ3 (see ref. [21], whose structure is already known) on top of penetration protein μ1, account for only about 2.4% of the total particles in our sample, limiting the resolution of our virion reconstruction to 8.6 Å (Supplementary Fig. 2); thus, a higher resolution reconstruction was not pursued further.

Within the inner capsid shell of the final asymmetric reconstruction of the ISVP, two kinds of densities can be distinguished: filamentous, located inside the capsid; and non-filamentous, occupying local regions underneath ten of the twelve icosahedral vertices (Fig. 1b–d and Supplementary Movie 1). In particular, when examined at a higher density threshold, the filamentous densities have the defining features of dsRNA: right-handed duplexes with major and minor grooves (inset of Fig. 1b). Therefore, we attribute the filamentous densities to viral RNA and the non-filamentous densities to proteins—more specifically, TEC, surrounded by the dsRNA to be transcribed.

MRV contains ten TECs, each under one of its twelve vertices, as identified in the Earth-like Mercator projection (i.e., the surface of a sphere is projected into a rectangle map) of the densities inside the core of the asymmetric reconstruction (Fig. 1c). Building on the reference to the Earth, MRV's 3-fold axis is analogous to the Earth's axis of rotation and MRV's Mercator projection is analogous to the Earth's map. Accordingly, six of the ten TECs are located near the two poles (pole-proximal) and are related to each other by $D_{3d}$ symmetry; the remaining four are located near the tropic (tropic-proximal), two above and two below the tropic, and are related to each other by pseudo-$D_{3d}$ symmetry (Fig. 1c, e). The two vertices without a TEC, one above and one below the tropic (Fig. 1e), are occupied by dsRNA densities (red circles in Fig. 1c); consequently, three duplexes of dsRNA extend from the north pole to the south pole across these two unoccupied tropic-proximal vertices, in a right-handed

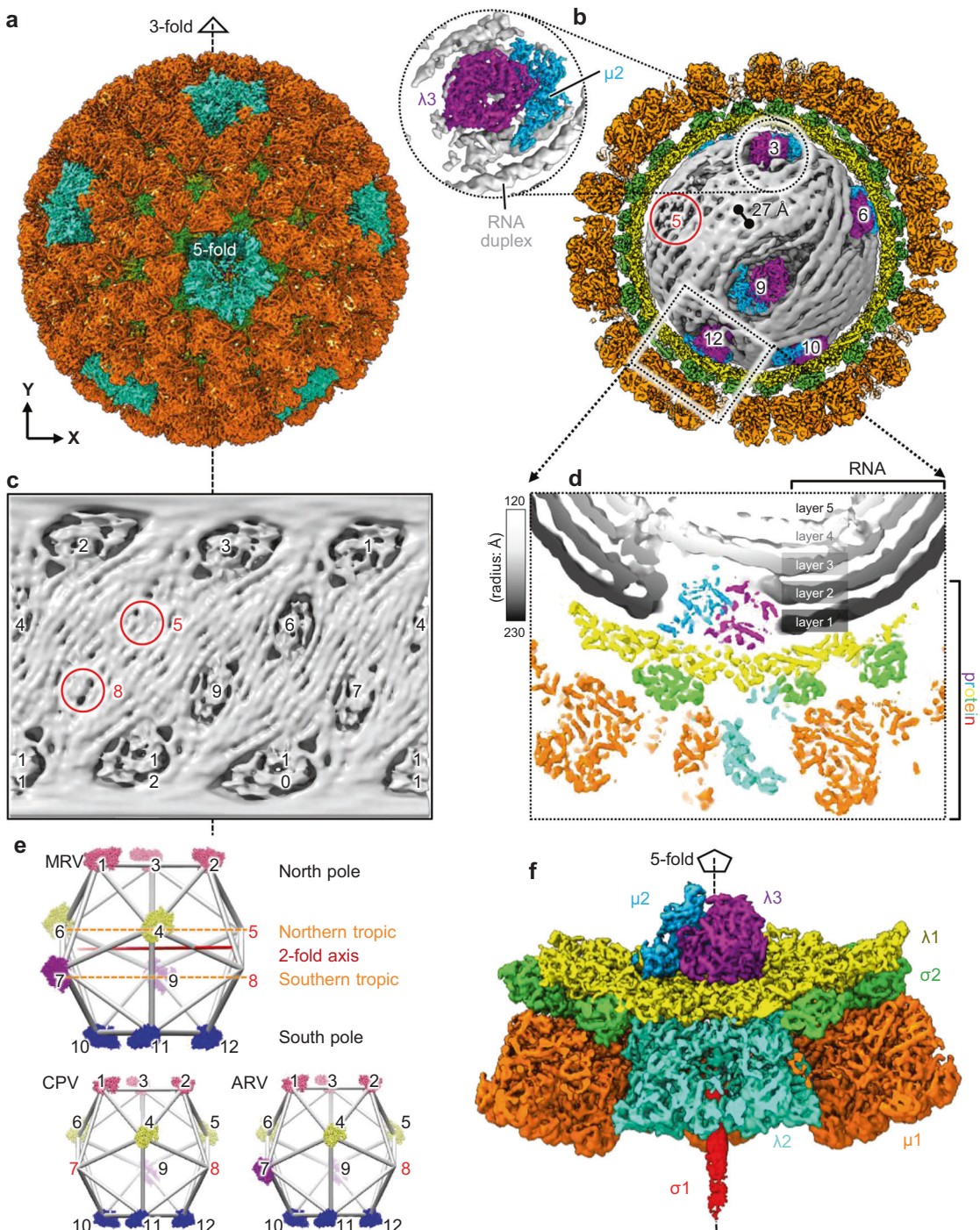

**Fig. 1 Transcription enzyme complex (TEC) and dsRNA organization revealed by asymmetric reconstruction of mammalian reovirus (MRV) infectious subvirion particle (ISVP). a** Surface representation of the asymmetric reconstruction of MRV ISVP. The **Y**-axis lies along a pseudo-3-fold symmetry axis. **b** Cut-open view of the capsid, exposing the RNA genome and TECs. The twelve vertices are numbered from 1 to 12 (not all shown). NTPases and RNA-dependent RNA polymerases (RdRps) colored in blue and purple, respectively, and RNA duplexes colored in gray. The red circle indicates the unoccupied 5-fold vertex. The inset (black circle) is the zoom-in view of a TEC and its surrounding RNA duplexes. **c** Mercator projection of the core density illustrating the organization of ten TECs. The two TEC-absent vertices (red circles at positions 5 and 8) are occupied by three RNA duplexes. **d** Slice view of the boxed region in (**b**), highlighting the five concentric layers of RNA duplexes and virus proteins. **e** Illustration of TEC organization of MRV, cytoplasmic polyhedrosis virus (CPV), and aquareovirus (ARV). The vertices are aligned for ease of comparison. The TEC-absent vertices are highlighted in red. Particle sizes are not to scale. **f** Surface representation of the sub-particle reconstruction of the TEC-containing vertex, with same view and colors as (**d**). Vertices are numbered the same way in (**b**, **c**, **e**).

fashion around the 3-fold axis (Fig. 1c). The dsRNA orientation set by the unoccupied vertices may contribute to the overall right-handed pattern of the entire genome (Fig. 1b, c). There are seven right-handed RNA duplexes between the two tropic-proximal TECs on the same latitude, e.g., between TEC 4 and 6, and the two RNA duplexes between two tropic-proximal TECs across the equator, e.g., between TEC 4 and 7 (Fig. 1c).

Among the many viruses within the *Spinoreovirinae* subfamily of the *Reoviridae*, only MRV (this study), CPV[22,23], aquareovirus (ARV)[24], and Fako virus (FAKV)[25] have their TEC organizations resolved to date. The TEC organizations of MRV, CPV, and ARV were resolved by classification of symmetry-expanded particles without alignment, while the TEC organizations of FAKV was based on probabilities estimated by cross-correlating with various arrangements of TEC decoys[25]. Though their TECs all follow pseudo-$D_{3d}$ symmetry[22,24], the locations of their unoccupied vertices within the capsid differ (Fig. 1e). While MRV and CPV both have ten TECs in each capsid, ARV has eleven, and FAKV has nine. All of their six pole-proximal vertices are occupied by TECs. However, this is not the case for their six tropic-proximal vertices: there is only one unoccupied vertex in ARV, while two unoccupied vertices in MRV, CPV, and FAKV: one on either side of the equator in MRV (vertex 5 and 8) but both on the same side of the equator in CPV and FAKV (vertex 7 and 8). In other words, in terms of their TEC organizations, CPV, ARV, and FAKV have polarity along 3-fold axis, but MRV does not. Except for the multiple TEC organization possibilities suggested for FAKV[25], a common trend among the viruses with unique TEC organization is that the number of genomic dsRNA segments of these viruses matches the number of TECs within each of their capsid.

Visible dsRNA density in the asymmetric reconstruction of the ISVP includes five concentric layers, layer 1–5, located at particle radii of 230, 209, 186, 161, and 139 Å (Fig. 1d), respectively. The distance between adjacent RNA duplexes within layer 1 is 27 Å (Fig. 1b). Taken together, our asymmetric ISVP and sub-particle reconstructions show that each MRV ISVP contains, from inside out, 10 TECs (a heterodimer of λ3 and μ2, Fig. 1f), 120 copies of CSP λ1, 150 copies of clamping protein σ2, 60 copies of mRNA capping protein λ2 which form the 12 pentameric turrets, 600 copies (or 200 trimers) each of the membrane penetration protein μ1 and its protection protein σ3, and 12 trimers of receptor binding protein σ1 [visible only in the sub-particle reconstruction (Fig. 1f) for reasons described in the "Methods" section].

**RdRp cofactor μ2 is an NTPase and has a dynamic RNA-binding domain**. Anchored to each occupied vertex is a TEC, which is a heterodimer composed of one RdRp λ3 and its co-factor μ2 (Figs. 1f and 2a) and surrounded by genomic dsRNA (Fig. 1b). Our vertex sub-particle reconstruction at 3.3 Å resolution revealed amino-acid side chains that helped building the atomic model of μ2 de novo (Fig. 2, Supplementary Fig. 4 and Supplementary Movie 2). Within μ2, we observed a density attributed to the γ-phosphate group detached from NTP (inset of Fig. 2d), similar to that observed in ARV NTPase VP4 (Supplementary Fig. 5) (see ref. [24]), indicating that MRV μ2 is also an NTPase (Supplementary Fig. 5). We were able to model 680 of the 736 total amino acid residues of μ2 (Supplementary Fig. 6). The atomic model of μ2 has a triangular shape (Fig. 2b), comprising three domains: an N-terminal, RNA-binding domain[20] (RBD, residues 1–264, see details in the TEC–RNA interaction section below), an NTPase domain (residues 265–607) with a bound γ-phosphate group, and a C-terminal domain (CTD, residues 608–736) (Fig. 2c–e and Supplementary Movie 2).

The RBD contains eleven α helices (Rα1–11) and two parallel β sheets (one formed by strands R1, 2, 6, 7, and 8 and other by strands R3-5) (Fig. 2b, e). The RBD can be further divided into two subdomains—tip and main—according to their electrostatic potentials. The positively-charged tip subdomain, distal to the NTPase domain, contains α helices (Rα5–8) and loops, which connect to the main subdomain (Fig. 2b–e). The negatively-charged main subdomain of RBD, proximal to the NTPase domain, is a condensed structure of alternating α helices and β sheets that resembles the proximal region of ARV N-terminal domain (NTD). The comparison was limited to the proximal region because the distal region of the main subdomain, and the entire tip subdomain of ARV NTD could not be modeled due to low resolution density[24], indicating the region's high flexibility. In the main subdomain, sequences for two density regions were assigned but not modeled due to sub-optimal, low-resolution density quality: one (residue 190–196) on the inner surface of RBD with fragmented density and the other (residues 265–283) at the junction of RBD and NTPase domain with density only visible when low-pass filtered. The existence of these low-resolution features in the otherwise well-defined overall density of RBD suggests that RBD is a dynamic structure, consistent with prior observation that most of the homologous regions in ARV NTPase VP4 are highly flexible thus not modeled[24]. Moreover, the main subdomain's alternating α helices/β sheets and the two subdomains are connected by many loops, which may collectively confer flexibility to the overall RBD structure.

The NTPase domain contains twelve α helices (Nα1–12) and two antiparallel β sheets (one formed by strands N1 and 6 and other by strands N2–4) (Fig. 2d). At the center of the NTPase domain, the NTP binding site, comprising loop 410–414 and helices Nα10–11, holds the aforementioned γ-phosphate group (inset of Fig. 2d and Supplementary Fig. 5d), as also observed in ARV[24]. The NTPase domain is conserved among MRV, CPV, and ARV[22,24] and is the most stable domain in all three viruses, indicated by the well-defined density of this domain (Supplementary Fig. 4).

Extending from the NTPase domain is the CTD, which comprises two short α helices (Cα1 and 2), two antiparallel β sheets (one formed by strands C1, 2, and 5 and other by strands C3 and 4) (Fig. 2c), and three long loops pointing towards inner-layer RNA densities (Fig. 2a, b). The RNA-proximal densities at the middle regions of the three long loops are of insufficient resolution for atomic modeling but can be assigned to the residues 625–635 (negatively charged), 714–720 (positively charged), and 670–680 (negatively charged), based on the locations of their immediate upstream and downstream modeled residues.

**In situ structure of RdRp λ3 and its interactions with NTPase μ2**. Based on our vertex sub-particle reconstruction, we were also able to build an in situ atomic model of RdRp λ3. Following the nomenclature established in the crystal structures of the recombinant λ3 (see ref. [16]), our in situ structure comprises five major domains: N-terminal domain (residues 1–386), thumb (residues 793–901), fingers (residues 387–556 and 595–690), palm (residues 557–594 and 691–792), and C-terminal bracelet (residues 902–1267) (Fig. 2f–h and Supplementary Movie 2). As in the crystal structures[16], four channels can be readily identified in the in situ structure of λ3: one into the RdRp on the opposite side of the capsid, the RNA template entrance; one on the side of the capsid, the transcript exit; one at the middle of the bracelet, the template RNA exit; and one on the opposite side of the bracelet, the NTP entrance (Fig. 2h). Our in situ structure is more similar to the crystal structures of λ3 with two nucleotides (i.e., initiation

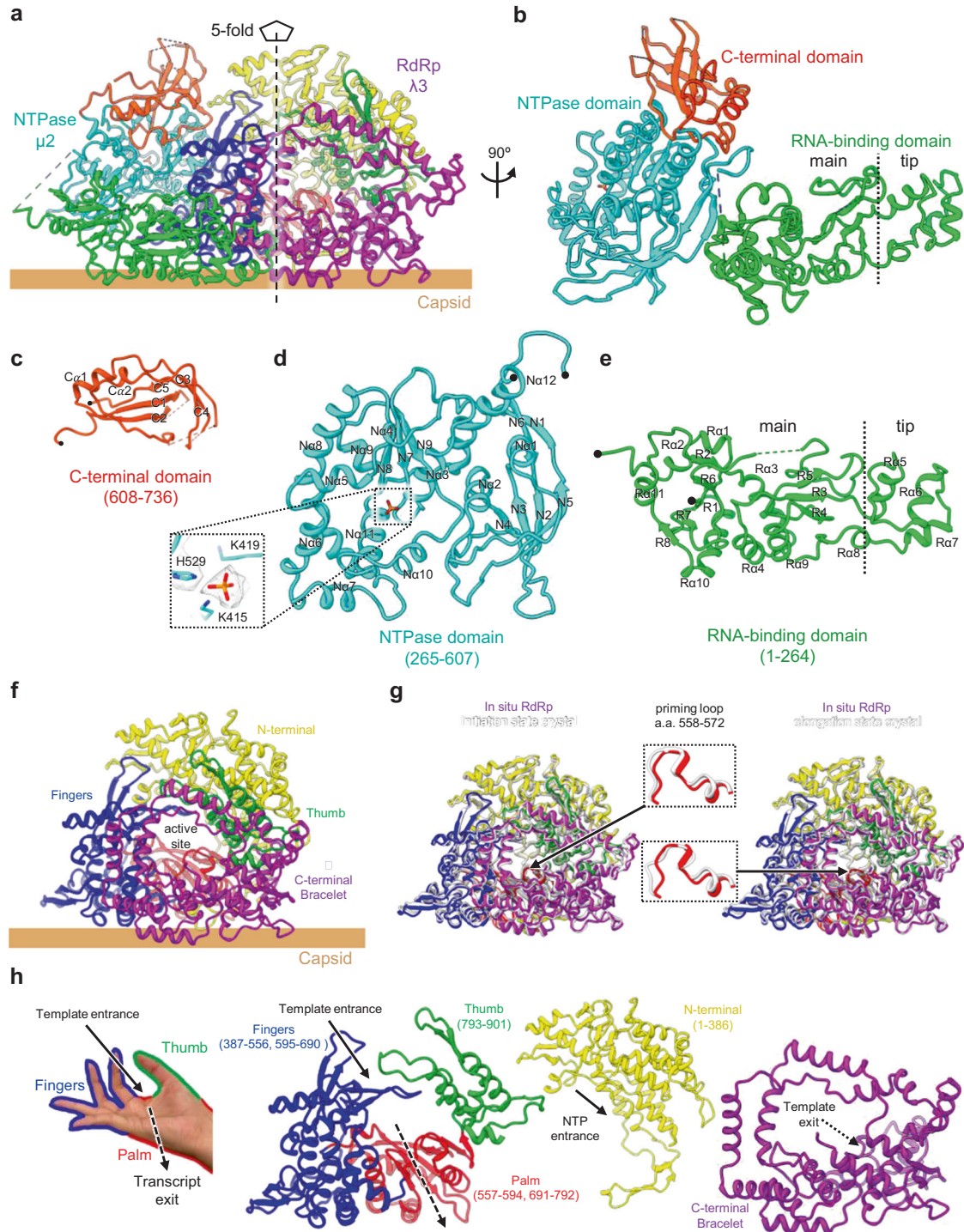

**Fig. 2 In situ structure of the TEC. a** Ribbon representation of TEC, which contains NTPase μ2 and RdRp λ3, with domains distinguished by different colors. **b** Ribbon representation of NTPase μ2 and its three domains, RNA-binding domain (RBD), in green, NTPase domain, in cyan, and C-terminal domain (CTD), in orange. The unmodelled residues are shown in dashed lines colored as domains. The vertical dotted line on the RBD is the divider between the "main" and "tip" subdomain. **c–e** Ribbon representation of μ2 domains: CTD (**c**), NTPase domain (**d**), and RBD (**e**), reoriented to provide a clear view of the secondary structure elements (labeled). The inset in (**d**) shows the atomic model (red sticks) and density (mesh) of the bound γ-phosphate group. **f** Ribbon representation of RdRp λ3 structure and its five domains: N-terminal domain (yellow), thumb (green), fingers (blue), palm (red), and C-terminal bracelet domain (purple). **g** Superpositions of the ribbon representations of our in situ cryoEM structure [colored as in **f**] and the crystal structure structures (gray) of λ3. **h** Right-hand illustration and ribbon representation of the RdRp domains, highlighting the entrances of NTP and template RNA as well as exits of RNA transcript and template RNA.

state, PDB: 1N1H) with an RMSD of 0.953 Å between the two structures, as compared to RMSD of 1.044 and 1.015 Å, between our in situ structure and the crystal structures of one and four phosphodiester bonds (i.e., elongation state, PDB: 1N38 and 1N35) (see ref. [16]), respectively (Fig. 2g). At the active site of our in situ structure, the priming loop (residues 558–565) extends from the palm domain, blocking the synthesis of double-stranded product, similar to that in the crystal structure of the initiation state (Fig. 2g). These two features suggest that our in situ structure is at the initiation state. Nonetheless, for both initial state structures, the in situ structure is a bit more spatially expanded than the crystal structure, as indicated by their total calculated solvent-accessible surface areas of 49,604 Å$^2$ versus 46,263 Å$^2$, and total volumes of $1.71 \times 10^5$ Å$^3$ versus $1.63 \times 10^5$ Å$^3$.

Aside from the significance as the atomic structure of transcription cofactor μ2, our in situ structure of TEC allows the exploration of the interface between our NTPase μ2 and RdRp λ3 to identify the various interactions between these two proteins (Fig. 3a, b). All three domains of μ2 interact with λ3, with a total contact area of approximately 745 Å$^2$. Residues lining the interface include those forming hydrogen bonds (Fig. 3c, d) and those conducive to the hydrophobic effect (Fig. 3e, f). For example, hydrogen bonds form between N536 of the μ2 NTPase domain and S392 of the λ3 fingers domain and between N690 of the μ2 CTD and E468 of the λ3 finger domain (Fig. 3d). By contrast, interactions between the μ2 RBD and the λ3 fingers domain include both hydrogen bonds (between Q236 of μ2 and A638 of λ3, Fig. 3d) and hydrophobic effects (between F230 of μ2 and I626 of λ3, Fig. 3f).

**Interactions between TEC and RNA.** Many interactions can be identified between dsRNA densities and multiple regions of TEC (Fig. 4). Based on clearly visible major and minor grooves of these RNA densities (inset of Fig. 1b and Supplementary Movie 3), backbone poly-AU duplex models were built for four dsRNA fragments around the TEC (Fig. 4a, b). All these RNA duplexes interact through hydrogen bonds with regions of TEC, including the μ2 RDB tip subdomain, NTPase domain, RdRp NTD (Fig. 4c–g). In addition, we identify a mechanism for RNA to cross different layers of the genome: on the "front-mid" RNA duplex, the end opposite to the hydrogen bond interaction faces many negatively charged residues on RBD main subdomain, repelling the negatively charged RNA duplex to cross from layer 1 to layer 2 (Supplementary Fig. 7a, b); likewise, on the "back-top" RNA duplex, the end opposite to the hydrogen bond interaction faces many negatively charged residues on the μ2 NTPase domain (Supplementary Fig. 7c), repelling the RNA duplex to cross from layer 2 to layer 3.

Further exploration into the interaction between RNA and TEC led to the identification of a branch point of the RNA duplex near the RdRp's template-RNA entrance (Fig. 4h). We interpreted this branch point to be the transcription/replication fork of the terminal dsRNA (Fig. 4h, i) and the "front-mid" RNA duplex to be the tail portion of the same genomic RNA segment, as also visualized inside CPV at the transcription initiation state[23]. Therefore, the RNA branch located at the template entrance of the RdRp is the 3′ end of the negative-strand RNA template and the other branch is the 5′ end of the positive-strand RNA non-template (Fig. 4h and Supplementary Movie 4).

In our structure, the template RNA has already passed through the positively charged entry tunnel to reach the RdRp active site (Fig. 4j). Though we did not build a model for the terminal RNA, the μ2 CTD interacts with the terminal RNA to guide the branching and entrance to the entry tunnel (Fig. 4j), consistent

with earlier suggestions of μ2 as a helicase[26]. At the active site near the template RNA is a density that appears to be the putative transcript (Fig. 4k, l). This putative transcript density is supported by residue S561 of the priming loop (Fig. 4l), which is known to regulate the progression of RNA transcription from the initiation to the elongation state[16,27]. The non-template RNA is guided by the positively charged surface from the RdRp C-terminal bracelet to the edge of the template exit (Fig. 4i and Supplementary Fig. 7d), thus positioned to reanneal with the exiting template RNA into a duplex. These structural observations are consistent with our aforementioned RdRp model being at the transcription initiation state[16].

**Loops and N-terminal fragments of CSP λ1 conformers are involved in TEC and RNA interactions.** Our asymmetric reconstruction resolved key elements of CSP λ1 interacting with TEC that were not previously resolved in the crystal structure of the MRV core with icosahedral symmetry imposed[13]. The crystal structure of the icosahedral core contains 60 identical λ1 homodimers. The two subunits within each homodimer have slightly different conformations: conformers λ1A near the 5-fold axis and λ1B near the 3-fold axis. Our asymmetric vertex sub-particle reconstruction now further reveals subtle structural differences among the five λ1A/λ1B dimers surrounding each vertex, hereby designated as λ1A$_1$/λ1B$_1$ through λ1A$_5$/λ1B$_5$ (Fig. 5a, b). Under this convention, λ1A$_{1-2,5}$/λ1B$_{1-2,5}$ forms the primary platform to which the TEC binds; by contrast, λ1A$_{3-4}$/λ1B$_{3-4}$ make minimal contact, as predicted earlier[18].

The asymmetric structures resolved in the five conformers of λ1A are primarily the N-terminal fragments (Fig. 5b, c). The N-terminal fragment (residues 222–240) in conformer λ1A$_1$ is a long loop, extending from the λ3 fingers domain to μ2 NTPase domain (Fig. 5d). By contrast, the N-terminal fragment (residues 194–240) in conformer λ1A$_2$ is a loop-helix-loop structure, extending from the λ1 surface to the RdRp λ3 bracelet domain (Fig. 5e). For the λ1A$_3$ N-terminal fragment (residues 211–230), we identified a helix-loop structure near the λ1A$_2$ N-terminal fragment that extends from the RdRp λ3 NTD towards the bracelet domain (Fig. 5e). For the λ1A$_4$ and λ1A$_5$ N-terminal fragments (residues 214–226 and 213–225), we identified a short helix region extending toward the RdRp (Fig. 5f). When λ1A$_4$ and λ1A$_5$ are superimposed, these short helices almost completely overlap each other in both overall structure and relative position to Loop 581–590 (Fig. 5c). Notably, this N-terminal helix in the homologous CSP A$_2$ and A$_4$ subunits in rotavirus have been shown to be a transcriptional regulator[28–30], and similar RdRp interactions between MRV and rotavirus for this N-terminal fragment of λ1A$_2$ and λ1A$_4$ suggest a similar role. Taken together, these structures indicate that the role of N-terminal fragments of λ1A conformers is to anchor the TEC to the capsid and possibly also to regulate transcription.

Near the λ1 N-terminal fragments, we also identified two sets of loops with differing conformations among the five λ1 pairs. The first set of loops (residues 562–571), which we named promiscuous loops (PL) because of their involvement in interactions with different partners, are part of λ1B$_{1-5}$ (Fig. 5c): the PL of λ1$_1$ interacts with the RdRp bracelet domain (Fig. 5g, red box); the PL of λ1$_2$ interacts with the back-mid RNA (Fig. 5g, orange box); the PL of λ1B$_3$ is not involved in any interactions (Fig. 5g, yellow box); the PL of λ1$_4$ sits underneath the front-bottom RNA, 5.9 Å away (Fig. 5g, green box); and the PL of λ1$_5$ interacts with the tip subdomain of NTPase RBD (Fig. 5g, blue box). The second set of loops (loop 581–590) are part of λ1A$_{1-5}$. When superimposed together, these loops align well, except for its

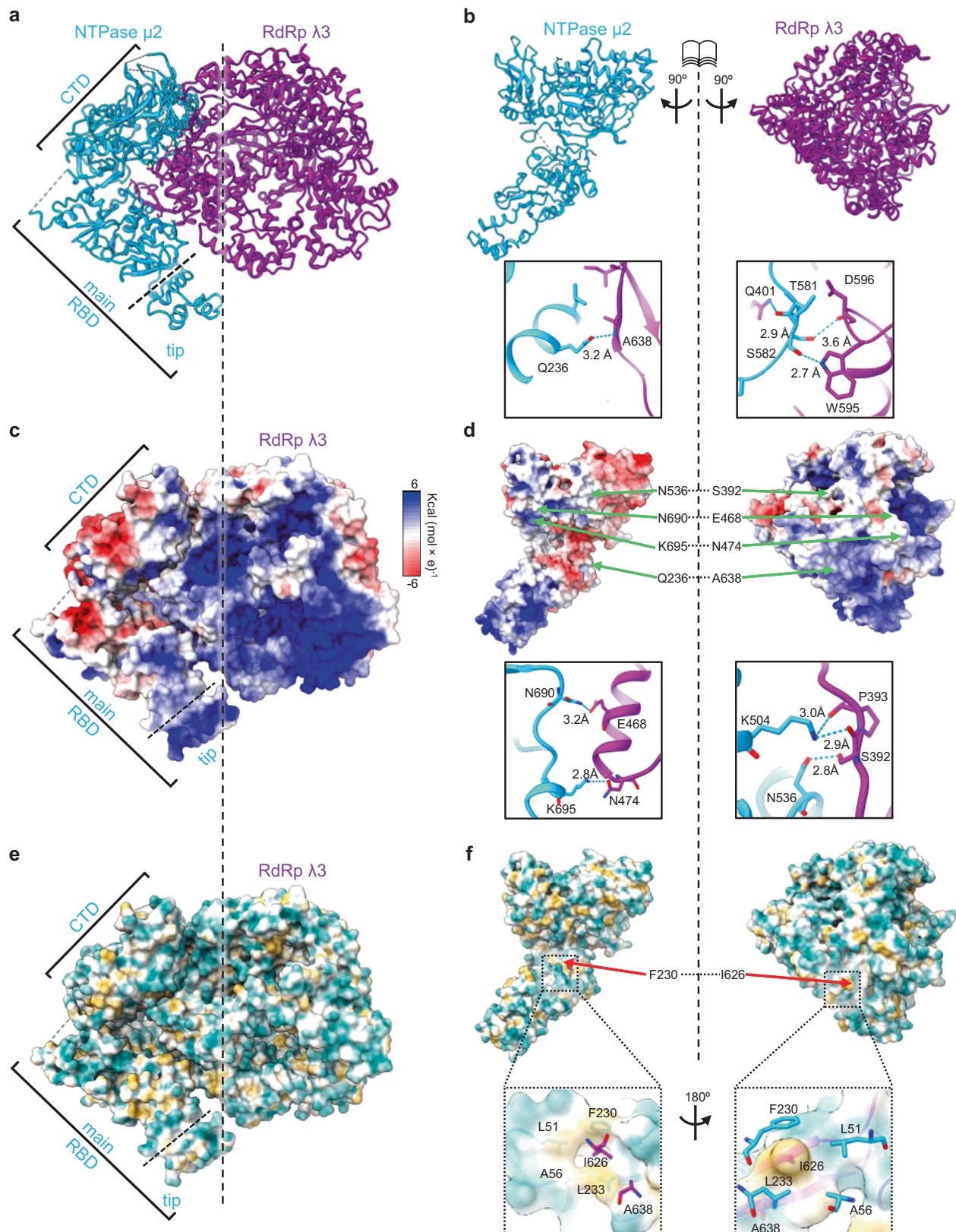

**Fig. 3 The interface between RdRp λ3 and NTPase μ2 and identification of key interacting residues. a**, **b** Ribbon representation of TEC (**a**), and NTPase μ2 and RdRp λ3 rotated 90° in opposite directions, exposing the interface and the interacting residues between the two proteins (**b**). **c**, **d** Same view as (**a**, **b**), with positive, neutral, and negative electrostatic potentials indicated in blue, white, and red, respectively. **e**, **f** Same view as (**a**, **b**), with hydrophilic, neutral, and hydrophobic area, indicated in cyan, white, and gold, respectively.

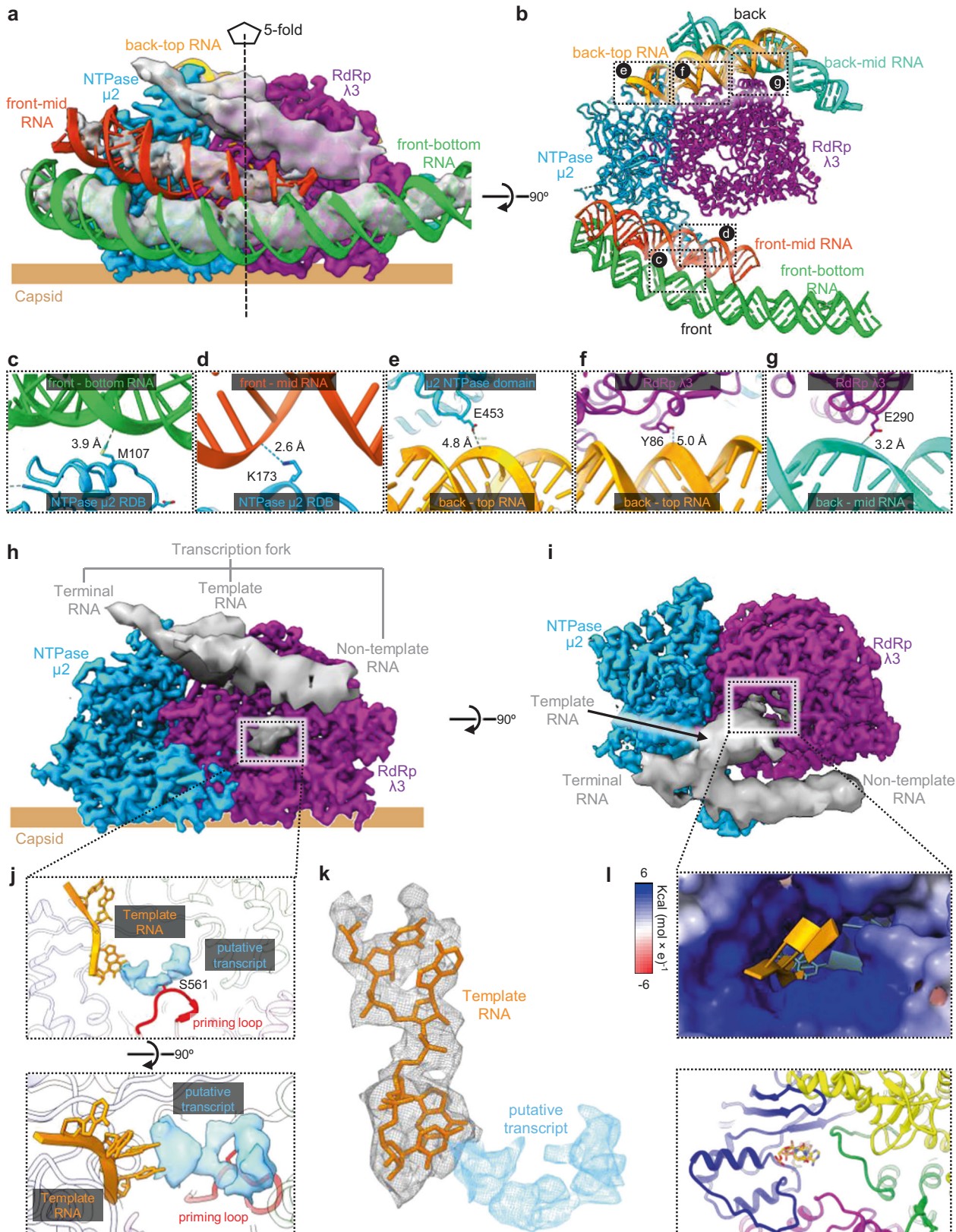

central region (Fig. 5c), to which vastly varying N-terminal fragments are attached, though not visualized.

**Loops in penetration protein μ1 contribute to virion assembly.** Previous efforts have fitted existing crystal structures[13,14] to low-resolution icosahedral reconstructions to suggest their interactions with the underlying inner capsid and with the capping enzyme[18,31]. Here, our near-atomic resolution structures reveal the atomic details of these interactions. The triangle in Fig. 6a demarcates an asymmetric unit of the icosahedral ISVP. Each asymmetric unit is composed of three and one third μ1 trimers—

**Fig. 4 Molecular interactions between TEC and RNA. a** Surface representation of TEC, with surrounding RNA strands in surface representation and duplexes in both surface and ribbon representation. **b** 90° rotated view of (**a**), with ribbon representation of TEC and RNA duplexes. **c**–**g** Zoom-in views of the boxed regions in (**b**), detailing hydrogen bond interactions between TEC and surrounding RNA duplexes. **h**, **i** Surface representation of TEC and unwound terminal dsRNA with transcription fork (**h**), and the 90° rotated view (**i**). **j** The zoom-in view of the boxed region in (**h**), showing the atomic models of initial template RNA and priming loop and the density of putative transcript within the RdRp active site, with RdRp in the ribbon representation and template RNA in stick representation. **k** Superposition of mesh and stick representation of the template RNA with mesh representation for the putative RNA transcript density. **l** Zoom-in view of the boxed region in **i**. Top panel, electrostatic surfaces representation of the RdRp, with positive, neutral and negative Coulomb potentials indicated in blue, white, and red, respectively. Bottom panel, ribbon representation of the RdRp and stick representation of the template RNA.

Q, R, S, and 1/3T trimers—totaling ten µ1 monomers (Fig. 6a, b)[31]. Each µ1 monomer is myristoylated at its amino terminus and consists of four domains, among which domain I has an autolytic cleavage site between N42 and P43, 8 Å apart (Fig. 6c). Our asymmetric reconstruction reveals two types among the ten N-myristoyl fragments in each asymmetric unit: nine are at the interface between two µ1 trimers, and one is at the interface between µ1 and λ2 (Fig. 6d–f). Notably, the N-myristoyl group of the first type is kept within each respective µ1's hydrophobic pocket[14,32], a helix-loop-helix motif (residue M212–R243) (Fig. 6e). In contrast, the N-myristoyl group of the second type is exposed and extended to interact with λ2 (Fig. 6f).

Aside from the µ1 N-terminal fragments, we also identified multiple conformations of µ1 Domain I at the interface with its neighboring µ1 trimers (Fig. 6g–i). The different conformations arise from the different positioning of the interacting loops of Domain I—namely, Loop 51–61 and Loop 72–96, which face toward the virus core. When viewed from inside out, each µ1 trimer is triangular (Fig. 6g), with three sides that interact with the neighboring proteins, each side harboring a Loop 51–61 and a Loop 72–96. The bond is formed via complementarity between Loop 51–61 of one µ1 trimer and Loop 72–96 of its neighboring µ1 trimer (Fig. 6g).

Based on the positioning of the interacting loops, two conformations between Loop 51–61 and Loop 72–96 can be identified. The first conformation, occurring between Loop 51–61 of trimer S and Loop 72–96 of trimer R (Fig. 6h), arises from lack of interaction due to the great distance between the interacting loops. Without interaction, Loop 72–96 of trimer R becomes flexible, hence its residues 81–90 are unresolved in our structure (Fig. 6h). The second conformation, which occurs between all other loops, arises from interactions between the two trimers (Fig. 6i).

**Interactions between symmetry-mismatched pentamer of capping enzyme λ2 and trimer of receptor binding protein σ1.** Five λ2 subunits form a pentameric turret with an axial channel (Fig. 7). In our structure of ISVP, loop 581–588 of λ2 bends towards the 5-fold axis (green in Fig. 7d, e) to avoid steric clash with N-myristoyl fragments of µ1 (Fig. 7c), unlike the unbent loop (gray structures in Fig. 7d, e) in the crystal structure of the core, in which µ1 is absent[13]. Extending from the axial channel of the λ2 turret is the trimeric σ1 receptor-binding spike[2,5,15], like many receptor-binding spikes in other viruses. Our 3D classification yielded two distinct sub-particle reconstruction classes of the λ2 turret: one (59.1%) with σ1 (Fig. 7a) and the other (40.9%) without (Fig. 7b). Fitting the σ1 trimer and λ2 crystal structures[13,15] to these reconstruction classes suggested how σ1 and λ2 could possibly interact. In this provisional fitting, the five D1283 residues from the λ2 subunits of the turret would form the vertices of a pentagon (Fig. 7c, f). Three σ1 subunits form a filamentous coiled-coil helix bundle projecting radially (Fig. 7a, c and Supplementary Movie 5) with their R48 residues projecting laterally to form the vertices of a triangle (Fig. 7f, g). Two of the three R48 residues of the σ1 trimer would be close enough to

create hydrogen bonds with their corresponding residue D1283 of λ2 subunits of the surrounding turret (Fig. 7f, g). As a result, there would be a free λ2 D1283 residue between the two aforementioned hydrogen bonds. The third R48 residue of the σ1 trimer would lie between the two remaining D1283 residues in the λ2 pentamer (Fig. 7f). Overall, as illustrated in Fig. 7g, such symmetry-mismatched positioning of the trimeric σ1 inside the axial channel of a pentameric λ2, and the resulting interaction thereof, could balance the competing needs of the σ1 spike. This arrangement would allow the spike to first associate with, and then dissociate from, the turret during capsid assembly and viral infection, respectively.

**Discussion**

The work reported here represents the atomic description of an infectious MRV particle. MRV exhibits both similarities and differences in its genome and TEC organizations when compared to ARV and CPV, two other members with known structures[22,24] of the *Spinoreovirinae* subfamily of the *Reoviridae* family. The genome of both MRV and CPV is segmented into ten segments[12,33] and of ARV into eleven[34]. Within each of these three viruses, the number of dsRNA segments equals the number of their TECs (Fig. 1) (see refs. [22,24]). Remarkably, the organizations of the TECs differ among these viruses (Fig. 1e). Their genome segments can be classified into three groups based on the length of each dsRNA segment: large, medium, and small (Supplementary Table 2). MRV and ARV both contain three large and three medium segments but four and five small segments, respectively[12,35]. CPV contains four large, two medium and four short segments[8] (Supplementary Table 2). Across all three viruses, each dsRNA segment encodes for one and only one specific protein[36,37] (Supplementary Table 2). With the µ2 structure presented here, the atomic structures of all these proteins are now available and structural comparison among them show that only four are structurally homologous and indeed functionally related: three proteins (CSP, RdRp, and the turret protein) encoded by three large segments, and one (the NTPase) encoded by one medium segment[38]. Therefore, these viruses' genomes have diverged substantially to allow incorporation of segments encoding for completely different proteins needed to interact with different host cells, as exemplified by the proteins involved in cell entry (µ1 for MRV and spike protein for CPV, e.g., Supplementary Table 2). In addition, it remains controversial regarding possible bias in the various approaches used for decoupling asymmetric structures from the icosahedral arranged components. Indeed, the TEC organizations of Fako virus were not unique based on probabilities estimated by cross-correlating with various arrangements of TEC decoys[25], and those within non-turreted dsRNA viruses[28,39] in the *Sedoreovirinae* subfamily of the *Reoviridae* are yet to be established. Because many mechanisms, including but not limited to recombination, duplication and hyper-mutation and overprinting, could have given rise to the conserved untranslated ends among RNA segments of these viruses, it remains a mystery how the different TEC organizations reported here (Fig. 1e) have emerged.

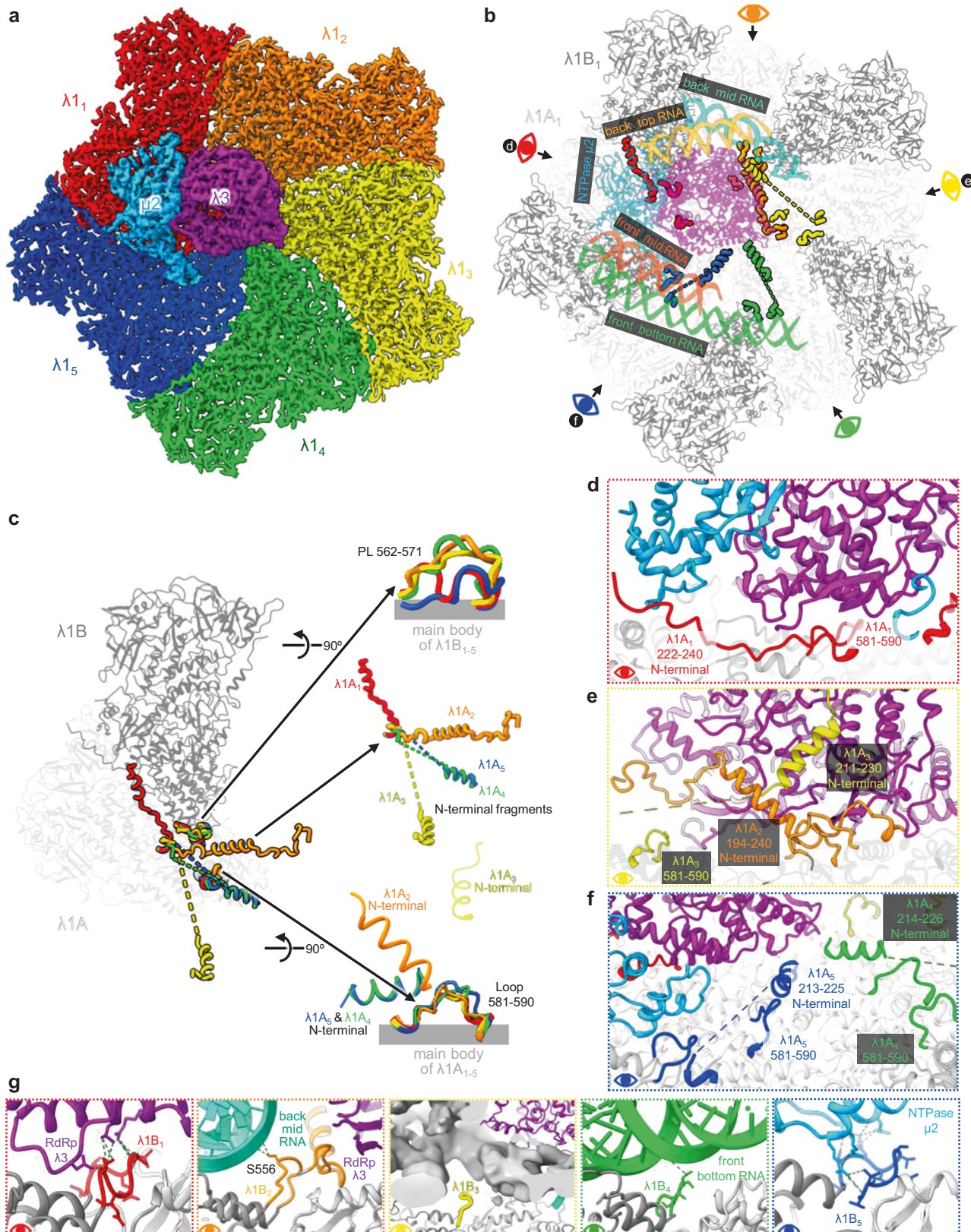

**Fig. 5 N-terminal fragments and loops of λ1 involved in TEC and RNA interactions. a** Surface representation of λ1$_{1-5}$, colored in red, orange, yellow, green, and blue respectively, and TEC, colored as in Fig. 4, viewed from inside of ISVP. **b** Ribbon representation of λ1, TEC, and surrounding RNA duplexes. Main bodies of λ1A$_{1-5}$ and λ1B$_{1-5}$ represented in light and dark gray, respectively. Loops 581–590 of λ1A$_{1-5}$, the N-terminal fragment of λ1A$_{1-5}$, and promiscuous loops (PL) 562–571 of λ1B$_{1-5}$ are colored as in (**a**). Eye symbols indicate different view directions. **c** Superposition of ribbon representation of λ1$_{1-5}$ pairs for easy comparison of different conformations, colored as in (**b**). **d–f** Ribbon representation of N-terminal fragments of λ1A$_{1-5}$ from the view indicated by the eye symbols in (**b**). **g** Ribbon representation of promiscuous loops of λ1B$_{1-5}$ from the view indicated by the eye symbols in (**b**).

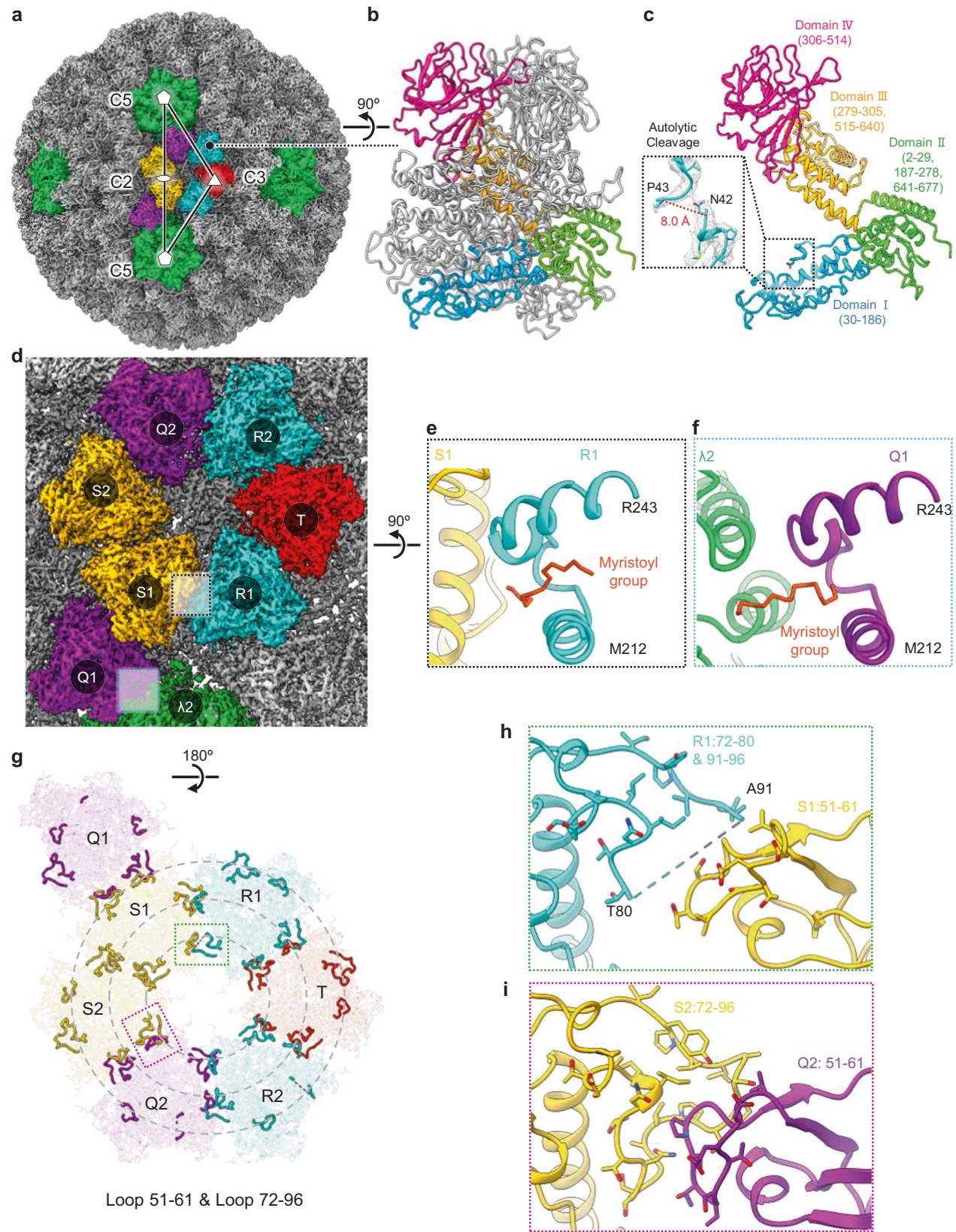

Loop 51-61 & Loop 72-96

Among the atomic models of the aforementioned four MRV proteins with homologs in both ARV and CPV[13,23], only that of the NTPase μ2 is built de novo in the current study. Our μ2 atomic model contains both conserved and non-conserved regions when compared to ARV NTPase VP4 and CPV NTPase VP4. Indeed, despite only 25% sequence identity match between MRV and ARV NTPases and no recognizable match between MRV and CPV NTPases at the sequence level, the NTPases from MRV, ARV, and CPV all share the same dihedral angle in NTPase domains. The similarities in their NTPase domains are reflected by the matching helices shaping their NTP binding sites (Supplementary Fig. 5d–f). By contrast, these

**Fig. 6 Autolytically cleaved penetration protein μ1 with myristoyl groups and interaction loops. a** Density map view of ISVP facing the 2-fold axis. λ2 is colored in green, while μ1.Q, S, R, and T trimers are colored in purple, yellow, blue, and red, respectively. White pentagons, triangle, and oval represent the icosahedral 5-fold, 3-fold, and 2-fold symmetry axes, respectively. Black triangle, connecting white symmetry points, represents an asymmetric unit. **b** Ribbon representation of μ1 trimer, with one μ1 monomer colored by domain (domain I, blue; domain II, green; domain III, yellow; domain IV, magenta). **c** Ribbon representation of isolated μ1 monomer, showing autolytic cleavage between residues 42 and 43 in the boxed region. **d** Zoom-in surface representation of μ1 hexagon from (**a**), formed by μ1 S1, S2, Q2, R2, T, and R1 trimers. **e** Zoom-in view of R1's myristoyl group (stick representation) and hydrophobic pocket (ribbon representation), as well as neighboring S1 (ribbon representation) from black box in (**d**). **f** Zoom-in view of Q1's myristoyl group (stick representation) and hydrophobic pocket (ribbon representation), as well as neighboring λ2 (ribbon representation) from blue box in (**d**). **g** Flipped view of (**d**), in ribbon representation, highlighting loops 51–61 and 72–96, which are divided by three concentric dashed circles. **h** Zoom-in ribbon representation of R1's loop 72–96 (81–90 unmodeled) and S1's loop 51–61, from green box in (**g**). **i** Zoom-in ribbon representation of S2's loop 72–96 and Q2's loop 51–61, from magenta box in (**g**).

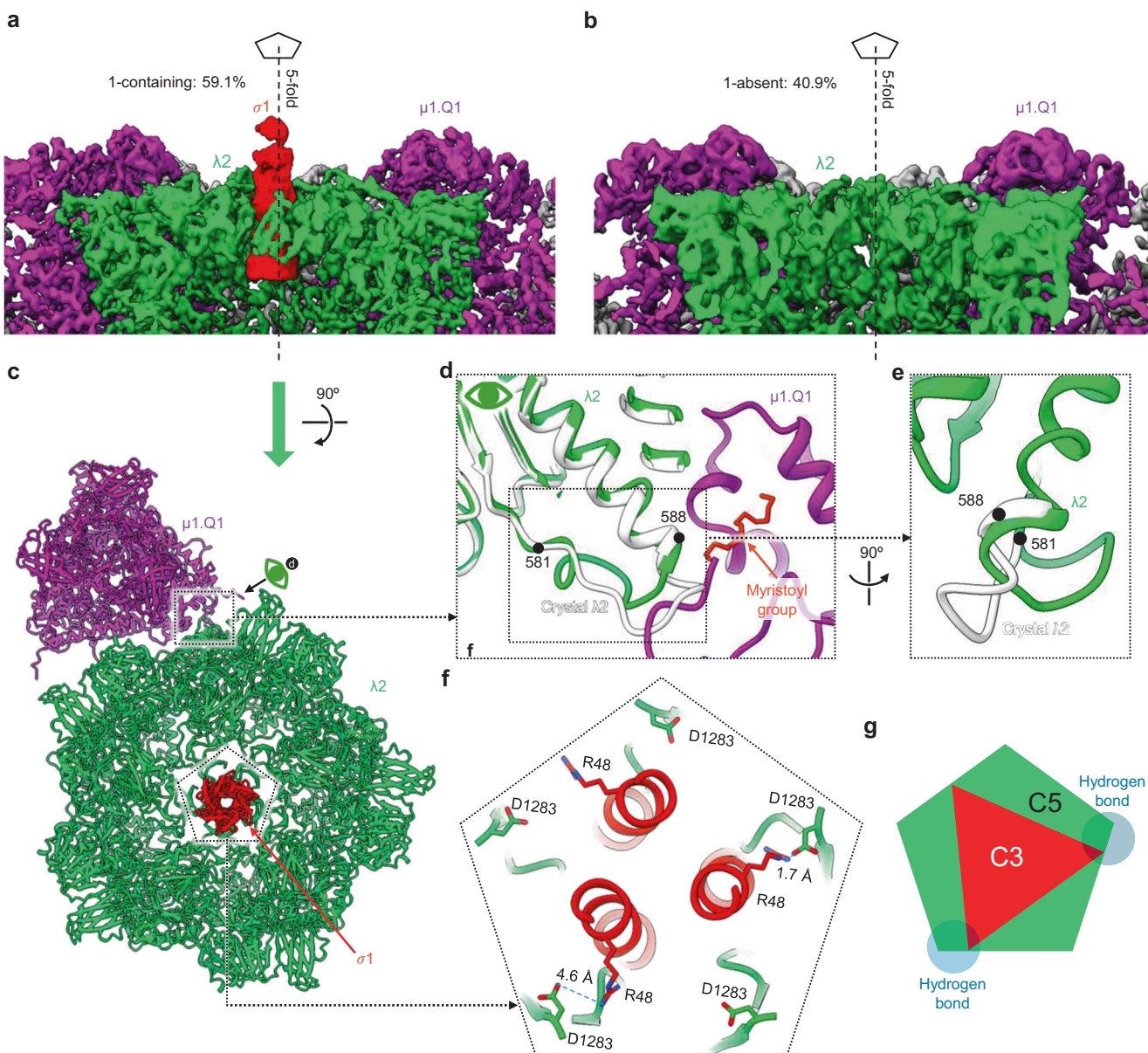

**Fig. 7 Hydrogen bonding between symmetry-mismatched receptor-binding protein σ1 trimer and capping enzyme λ2 pentamer. a**, **b** Surface representation of receptor binding protein σ1, mRNA capping protein λ2, and surrounding μ1. **c** Ribbon representation of σ1, λ2 and a surrounding μ1 trimer Q1 as viewed from the top of (**a**). **d** Superposition of the ribbon representation of our in situ λ2 structure (colored as in **a**) and X-ray λ2 structure (gray). The stick representation of Q1's myristoyl group and the ribbon representation of the hydrophobic pocket is also indicated. **e** Rotated zoom-in view of the dotted box in (**d**), highlighting the conformational difference of loop 581–588 between in situ and crystal structure. **f** Clipped view of the dotted pentagon in (**c**), detailing the interactions between σ1 and λ2. The interacting residues are represented in stick style. **g** Schematic illustration of symmetry mismatch between σ1 ($C_3$) and λ2 ($C_5$). The two hydrogen bond areas are indicated in transparent light blue.

homologous proteins are less similar in their CTD and NTD (in MRV, we show this to be an RNA-binding domain in the current study) (Supplementary Fig. 5a–c). Within their NTDs, the tip subdomains of ARV and CPV NTPases were poorly resolved and not modeled[22,24], indicating flexibility. The main subdomains of NTDs of MRV and ARV NTPases share similar positions of α-helices (Rα1–4, 10–11), β-sheet and unmodeled region (residue 190–196). However, within these main subdomains, the region that interfaces the NTPase domain is structurally unique to each virus; this region interacts with CSPs in different ways for each virus. Regarding the CTD, MRV μ2, and ARV VP4 are similar, while CPV VP4 lacks the CTD. MRV NTPase μ2 likely has additional roles as it possesses two N-terminal Pro residues (P204 and P208, respectively), which are known to support microtubule association and formation of inclusion bodies[40,41].

Across the three domains of the MRV NTPase μ2, both NTD (i.e., RBD) and CTD tend to be more flexible than the NTPase domain, which is highly conserved across different members of the *Spinoreovirinae* subfamily. RBD faces the template RNA exit, while CTD faces the terminal dsRNA entrance (Fig. 3h). We suggest that CTD guides the template strand of the terminal dsRNA into RdRp to initiate RNA transcription and the NTPase domain provides the required energy for this process, thus playing the role of a helicase, while RBD guides the template RNA out of the RdRp active site.

When TECs of ISVP are in the transcription initiation state, most of the N-myristoyl fragments of membrane penetration protein μ1 remain hidden inside the hydrophobic pockets (Fig. 6e). This observation suggests that the trigger for RNA transcription is not the attachment of μ1 to the cell membrane, but rather the removal of protection protein σ3 from the capsid surface[21]. During our sample purification, most σ3 has inadvertently detached from most virion particles (Supplementary Fig. 2), exposing membrane penetration protein μ1 and triggering its automatic cleavage (Fig. 6). The cleaved C-terminal fragments of μ1 could enter inside the capsid[10] thus relaying a message of cleavage (in a real, in vivo infection event, this would mean attachment to, and penetration through, the endosomal membrane, immediately prior to entering the host cytoplasm) to start RNA transcription. It is conceivable that only residual NTPs from the cell culture medium were present in the extracellular medium to "fuel" RNA transcription, halting ISVPs' transcription at the initiation state as observed. Lacking an external protein layer, the single-shelled CPV uses a different triggering mechanism[42]. The binding of SAM and GTP to the turret protein triggers a cascade of allosteric conformational changes, from the turret protein to the CSP, which in turn, changes the conformation of RdRp inside to shift from the meta-stable quiescent to the lower energy level active state[23,43]. Therefore, beyond the conserved mechanism of RNA transcription within the active site of the ancient hand-shaped core domain of RdRp, the triggering, regulatory, and organizational aspects of the endogenous RNA transcription of these dsRNA viruses have apparently diverged significantly, consistent with the highly mutable nature of RNA genomes and these viruses' need to adapt to their vast host ranges.

## Methods

**Viral culture and isolation**. Mammalian reovirus serotype Dearing 3 (ATCC, VR-824) was propagated in LLC-MK2 derivative cells (ATCC, CCL-7.1). Cells were maintained in Eagle's Minimum Essential Medium (EMEM) (ATCC, 30-2003) supplemented with 10% Fetal Bovine Serum (FBS) (Omega Scientific, FB-11), 100 I.U. penicillin and 100 μg/ml streptomycin (Corning, 30-002-Cl) and incubated at 37 °C and 5% $CO_2$ in air atmosphere of >95% humidity. Infection was performed in 18 aliquots of 38 ml culture flasks when cell culture reached 80% confluence. At 3–4 days of post-infection, when cytopathic effect began, supernatant (~700 ml) was collected after centrifugation at 10,000 × $g$ for 20 min to remove cellular debris. The clarified supernatant was centrifuged at 80,000 × $g$ for 1 h to pellet the virion

particles. Virion particle pellets were then resuspended in 10 ml PBS, pH 7.4, loaded onto a double sucrose cushion (50% wt/wt sucrose and 80% wt/wt sucrose, both in PBS, pH 7.4) and centrifuged at 100,000 × $g$ for 1 h. The band between the two sucrose cushions was collected, diluted 20-fold in PBS, pH 7.4, and centrifuged at 100,000 × $g$ for 1 h. The resulting pellet, which consisted mainly of virus particles, was resuspended in 20 μl of PBS, pH 7.4, evaluated for concentration and particle integrity using negative-stain (2% uranyl acetate) transmission electron microscopy, and subsequently used for cryoEM sample preparation.

**cryoEM**. For cryoEM grid preparation, 2.5-μl aliquots of the purified MRV sample were applied to a glow discharged Quantifoil R2/1 300-mesh holey carbon grid (Quantifoil, Micro Tools GmbH), manually blotted with filter paper (Whatman grade 1 from GE Healthcare), and plunged into a 60–40 propane–ethane mixture.

The grids were loaded into a Titan Krios electron microscope (Thermo Fisher Scientific) equipped with a Gatan imaging filter (GIF), and cryoEM images were recorded on a post-GIF Gatan K2 Summit direct electron detection camera operated in super-resolution electron-counting mode. The magnification was ×130,000, giving a pixel size of 1.07 Å/pixel at the specimen level. Data collection was facilitated by SerialEM[44]. The dosage rate was set to 56 electrons/Å² on the sample level, and the exposure time for each frame was 0.2 s. Targeted under-focus value was 1.8–2.2 μm. In total, 2032 movies were recorded.

**Single-particle icosahedral reconstruction**. We followed a workflow in the framework of Relion 3.0 (see ref. [45]) to carry out single-particle reconstruction of MRV. Frames in each movie were aligned and averaged with UCSF MotionCor2[46], generating two motion-corrected micrographs from each movie for distinct usages: a dose-weighted micrograph with high-resolution but low contrast for final reconstruction, and a non-dose weighted micrograph with high contrast but low-resolution for particle picking and estimation of the micrograph's contrast transfer function (CTF). Defocus values of micrographs were determined by CTFFIND4.1 (see ref. [47]). Micrographs with ice contamination were discarded, and the remaining micrographs were modified using EMAN2 (see ref. [48]), to prepare for the subsequent particle picking in Ethan[49], as detailed in the following: EMAN2 e2proc2d. py tool[48] was first used to rescale the non-dose weighted micrographs down to bin4, followed by low-pass filtering with frequency cutoff value of 0.2 Å$^{-1}$. Subsequently, we used Ethan to automatically pick circular particles with a radius of 65 pixels (i.e., 278.2 Å) from the EMAN2-processed micrographs. The Ethan program picked 11,229 particles and identified 43 micrographs with no identifiable particles that were discarded. Subsequently, from the same set of micrographs used in Ethan, we manually picked 12,122 additional particles with the "Manual picking" tool on the graphical user interface (GUI) of Relion (particle diameter = 800 Å). Both automatically and manually-picked particles were combined into a single Relion STAR file for each micrograph. We then extracted these particles using "Particle extraction" tool of the Relion GUI with the following parameters: box size: 1024 pixels, diameter background circle: 944 pixels, rescale size: 256 pixels. The extracted bin4 particles were classified with the "2D classification" tool of Relion GUI with the following parameters: number of classes: 100, regularization parameter: 2, number of iterations: 25, mask diameter: 1000 Å. After 2D classification, 11 "good" class averages exhibiting fine capsid features, comprising 14,465 particles, were manually selected for 3D auto-refine, using "Subset selection" tool on the Relion GUI. Particle images belonging to the selected "good" class averages were subjected to an icosahedral (option I3) refinement using "3D auto-refine" on the Relion GUI, resulting in a 4.3 Å resolution reconstruction. In this icosahedral reconstruction, we observed a faint outer density around the capsid protein μ1. To identify the faint outer density, we used Gaussian filter (width 2) in UCSF Chimera[50] to generate a low-resolution reconstruction, from which we were able to confirm that the faint densities are contributed by capsid protection protein σ3. This finding indicates that our 4.3 Å icosahedral reconstruction contains contributions from both ISVP and quiescent-state virions.

**Classifying and separating ISVP and virion particles**. In order to improve the homogeneity of particle populations, as indicated by the detection of faint σ3 density, we attempted to classify and separate the particles into two groups. However, the faint σ3 density present in the quiescent virion particles is not sufficient to drive a successful separation of the ISVPs and virion particles into different classes. To overcome this challenge, we subtracted density from the particle images to create more prominent differences between the ISVP and virion particles, prior to the 3D classification. The following details the workflow for achieving the separation of virion particles from ISVPs and subsequent final reconstructions (Supplementary Fig. 2). First, we generated capsid volume data for use in the subsequent mask creation step. To create the capsid volume data, we initially used the Volume Eraser function in Chimera to generate a sphere with 250 Å radius. Then, we moved this sphere to the center of the 4.3 Å reconstruction with the following command lines in Chimera: cofr #model_ID_of_4.3 Å reconstruction; move cofr mod #model_ID_of_volume_eraser. Finally, we erased the volume inside the sphere so that the outer ISVP capsid density and the outermost σ3 faint density remained in the post-subtraction volume, referred to as capsid volume. Second, we used this capsid volume as the input 3D map to generate a mask, using "Mask creation" tool on the Relion GUI. The output mask, when applied to the

4.3 Å reconstruction map, masked any remaining signal within it as 0. Third, we applied the mask to the particles that were used to build the 4.3 Å reconstruction map, using the "Particle subtraction" tool on the Relion GUI as follows: We used our 4.3 Å icosahedral reconstruction and its corresponding STAR file as "Map to projected" and "Input particles", respectively, and the mask from the previous step as "Mask to apply to this map" to generate the subtracted particles (Supplementary Fig. 2). Finally, the subtracted particles were subjected to a 3D classification step in Relion with icosahedral symmetry (option I3) into three classes, using the core density map as the reference map. (The core density was generated, following the same steps for creating the aforementioned capsid volume, except the volume outside the sphere was erased so that the core of the 4.3 Å icosahedral reconstruction remains.) This 3D classification step generated two identical core 3D class averages, representing the structure of ISVP, from 14,115 extracted particle images and one dissimilar class average, corresponding to the structure of quiescent virion, from 350 extracted particle images (Supplementary Fig. 2). ISVP-representing class averages were separated from quiescent virion class average using "Subset selection" tool on the Relion GUI. Particles belonging to either ISVP or quiescent virion class average were separately subjected to an icosahedral refinement (option I3) in "3D auto-refine" tool on Relion GUI, resulting in 3D reconstructions of ISVP and quiescent virion at resolutions of 4.3 Å and 8.6 Å, respectively (Supplementary Fig. 2). Only the particles in the ISVP group were subjected to further data processing.

**Vertex sub-particle reconstruction**. We followed a step-wise symmetry relaxation workflow (Supplementary Fig. 3) in the framework of Relion[45] to carry out asymmetric sub-particle reconstructions. First, the final 4.3 Å ISVP icosahedral reconstruction from above was symmetry-expanded with icosahedral symmetry (option I3), using *relion_particle_symmetry_expand* command, which generated an output Relion STAR file, containing 60 orientation entries for each ISVP particle. We then sorted these 60 entries for each particle into 12 groups, each group containing 5 entries with the same psi angle, i.e., entries belonging to the same vertex, and subsequently retained only the first of five symmetry-related occurrences in each vertex group in the final output Relion STAR file (using system command "sort −k 6,6 −k 4.1, 4.10 −u STAR file", where column six is _rlnImageName and column 4 is _rlnAnglePsi angle). Then, we extracted vertex sub-particles according to the STAR file from above, using the "Particle extraction" tool on Relion GUI with the following parameters: select the sorted STAR file from above as the input for the "Refined particles STAR file", set "re-center refined coordinates" to yes, set the "recenter on – **X, Y, Z** (pixel)" parameters as 0, 0, 114, set "Particle box size (pixel)" as 400 and "Re-scale size (pixel)" as 200. The output Relion STAR file was subjected to $C_5$ symmetry expansion using *relion_particle_symmetry_expand* command. In this resulting STAR file, $C_5$ symmetry-related orientations within each vertex share the same _rlnImageName. Subsequently, we conducted $C_1$ (asymmetric) 3D classification on the aforementioned $C_5$ symmetry-related sub-particles in Relion, with a reference map and a reference mask generated using Chimera, as detailed in the following work flow: First, we used "Volume Eraser" tool on Chimera to create and position a sphere on the 4.3 Å ISVP icosahedral reconstruction, according to the recenter coordinates and particle box size from the aforementioned sub-particle extraction process. More specifically, sphere radius of 200 pixels × 2.14 Å / pixel = 428 Å was moved to the recenter coordinates (0, 0, 114), which is 114 pixels × 2.14 Å / pixel = 244 Å away from the center of the 4.3 Å ISVP icosahedral reconstruction, along the positive **Z** axis. Second, the volume outside the aforementioned Volume-eraser sphere of 214 Å radius was erased and the remaining density was saved into a new mrc file. Finally, the new mrc file was rescaled using *relion_image_handler* command (-rescale_angpix 1.07 -new_box 400) and used as the reference map for the $C_1$ (asymmetric) 3D classification. As for the reference mask, we used the "Volume eraser" tool in UCSF Chimera to position a local spherical mask on the RdRp and NTPase region and finely adjusted the mask size to exclude as much of the capsid volume as possible. The $C_1$ (asymmetric) 3D classification step generated seven class averages, of which five classes displayed clear RdRp λ3 and NTPase μ2 structural elements. One of the aforementioned five class averages was selected using the "Subset selection" tool of Relion GUI for further processing as detailed in the following: We removed redundant sub-particles in the selected class average by examining the "_rlnImageName" and "_rlnMaxValueProbDistribution" of each sub-particle in the STAR file. More specifically, among the entries with the same "_rlnImageName" (column 6 in STAR file), i.e., within the same vertex, only one entry with the highest "_rlnMaxValueProbDistribution" (column 28 in STAR file) score was retained (using bash command "sort −k 6,6 −k 28.1,28.8 −u STAR file"). This process retained 102,966 sub-particles of good quality. Subsequently, we extracted these sub-particles without rescaling (bin1) using "Particle extraction" tool, followed by local refinement using "3D auto-refine" tool on Relion GUI, with the same reference map from the aforementioned 3D classification, which generated a 3.3 Å resolution asymmetric reconstruction of vertex sub-particle.

**$D_{3d}$-symmetry reconstruction of the ISVP core**. Under the Relion I3 convention, the **XZ** plane contains a 5-fold and a 2-fold axis with the **Z**-axis along the 5-fold axis, resulting in a total of four icosahedral vertices on the plane (two along the **Z**-axis and the other two away from it). We designated the position of the vertex sub-particle passing through the positive **Z**-axis as vertex 1 (V1) and the position of its

nearest vertex on the −**XZ** plane as vertex 2 (V2), as indicated in Supplementary Fig. 3.

Our 3.3 Å resolution asymmetric vertex map is reconstructed from ISVPs that share the same TEC orientations at one vertex. As per our designation rule from the previous paragraph, we designated this vertex as V1. Within this group of V1-aligned ISVPs, we sought for those that are also aligned at V2, in an attempt to reconstruct an asymmetric core map. To align the TEC orientation at V2, we followed the workflow as detailed below. First, we extracted V2 sub-particles from our V1-aligned ISVPs using the "Particle extraction" tool on the Relion GUI with the following parameters: select input file as data STAR file of the 3.3 Å asymmetric reconstruction, set "re-center refined coordinates" to yes, set the "recenter on − **X, Y, Z** (pixel)" parameters as −96, 0, −60, which are the **X** and **Z** vector components from V1 to V2, set "Particle box size (pixel)" as 400 and "Re-scale size (pixel)" as 100. The extracted V2 sub-particles (bin4) were subjected to 3D classification with a reference map generated by relion_reconstruct command (relion_reconstruct -i extracted V2 sub-particle.star -o reference map.mrc -ctf), a local spherical mask on the TEC region, and with no applied symmetry. This 3D classification generated three class averages, of which two classes displayed TEC structural density (Supplementary Fig. 3). We selected one of the TEC-displaying classes and removed redundant sub-particles in the selected class data STAR file, as described above. Subsequently, we extracted the bin2 ISVP core particles using the "Particle extraction" tool of Relion GUI with the following parameters: select the above STAR file as the input for "Refined particles STAR file", set "re-center refined coordinates" to yes, set the "recenter on – **X, Y, Z** (pixel)" parameters as 48, 0, −27, which are the **X** and **Z** vector components from V2 to ISVP core center, set "Particle box size (pixel)" as 1024 and "Re-scale size (pixel)" as 512. Finally, we ran relion_reconstruct command (relion_reconstruct -i extracted particles.star -o output name.mrc -ctf) without any symmetry, which generated a core reconstruction with twelve TECs that are related to each other by $D_{3d}$ symmetry. The twelve TECs were further divided into four groups, three in each group, according to their positions relative to the threefold axis: North Pole, Northern Tropical, Southern Tropical, and the South Pole.

**Decoupling $D_{3d}$ symmetry for asymmetric ($C_1$) reconstruction**. By adjusting the volume threshold of core reconstruction in Chimera, we discovered a cylinder "inner core", within the layers that embed twelve TECs, with a 3-fold axis of icosahedron symmetry along the centers of the cylinder bases, i.e., our core reconstruction contains $D_{3d}$ symmetry. Our next objective was to decouple $D_{3d}$ symmetry of the core and generate an asymmetric reconstruction to further examine the TECs at all vertices.

In order to decouple $D_{3d}$ symmetry of the ISVP core, we followed the work flow as described below. First, we generated a new STAR file (saved as rotated_core.star) that contains the rotated orientation parameters of the core particles so that the 3-fold axis is on the **Z**-axis. To accomplish this, we used relion_rotate_particles.py script from Localized Reconstruction[51] using the vector along the 3-fold axis (75.87, −211.3, 26.89) and data STAR file of the $D_{3d}$ core reconstruction (using the command line "python localrec-master/scripts/relion_rotate_particles.py -vector 75.87, −211.3, 26.89 -output rotated_core.star data.star"). Second, we generated a new rotated core map using rotated_core.star as the input of relion_reconstruct command (using the command line " relion_reconstruct -i rotated_core.star -o rotated_core.mrc -ctf"). The rotated_core.star was $D_{3d}$ symmetry expanded using relion_particle_symmetry_expand command. The symmetry-expanded output STAR file was then subjected to 3D classification without any symmetry ($C_1$), with the reference map (rotated_core.mrc) and a solvent mask. EMAN2_data python script was used to create the mask, such that the region excluding the core layer embedding twelve TECs was set to zero. This 3D classification step generated fifteen class averages, of which six classes showed reasonable and $D_{3d}$ symmetry-related core structures. The core structure had two vacant vertices, exhibiting pseudo-$D_{3d}$ symmetry among the remaining ten TECs. Subsequent 3D auto-refinement of one of six classes obtained a better resolution.

**Atomic model building and model refinement**. For atomic modeling, we followed an established protocol[52] described in the followings. We first fitted the existing crystal structures of MRV RdRp (PDB ID: 1MUK) and the cryoEM-derived model of ARV NTPase VP4 (PDB ID: 6M99) into our cryoEM density map of vertex, and identified the regions in the map that did not align well or were not modeled in the existing structures. MRV NTPase μ2 shares only 27% of sequence identity with ARV NTPase VP4, consistent with the observation that these two NTPases align only at some α-helix regions. As a result, MRV NTPase μ2 had to be modeled de novo. For other proteins, we first fitted the existing crystal structure into our cryoEM map and refined the fitted structure with the real space refinement tool in Phenix[53]. We then manually built atomic models from the regions that did not align well, using Coot de novo[54] as described in the following. First, we turned on the calculated map using the "Map Skeleton" tool. We used "Cα Baton Mode" tool in Coot to create Baton Atom by placing the Cα at the baton tip one by one, from N-terminus to the C-terminus along the map skeleton. Second, we converted baton atoms (i.e., Cα positions) into poly-alanine, main-chain model using the Coot's "Cα Zone to Mainchain" tool. Third, we mutated alanine residues to the specific amino acid residues of the MRV proteins with Coot's "Mutate Residue Range" tool by following the Cα density bumps and bulky side chain

densities (such as those of Tyr and Trp). Last, we performed local and global model refinements in PHENIX[53]. Poorly-fitting residues were manually adjusted using the Rotate/Translate, Rotamer Fit, and Real Space Refine Zone tools in Coot. Models were further refined using PHENIX's real space refinement tool[53].

**Modeling of dsRNA.** We generated a 7-bp ideal type A dsRNA model, using the "ideal DNA/RNA" tool in Coot, and fitted this ideal dsRNA model to the center region of the dsRNA density in the 3.3 Å resolution sub-particle reconstruction. Then, we extended from both ends of the ideal dsRNA model with Rosetta's custom functionalities[55].

**Model validation and visualization.** The quality of our final protein models was assessed, based on three areas: model geometry, fit to the density map, and agreement with the Ramachandran plot. Models were refined with PHENIX[53]. Electrostatics map was generated by PDB2PQR[56] and APBS[57]. Figures and movies were generated using UCSF Chimera[50] and Chimera X[58].

**Reporting summary.** Further information on experimental design is available in the Nature Research Reporting Summary linked to this paper.

## Data availability

The cryoEM density maps have been deposited in the Electron Microscopy Data Bank under accession codes EMD-31183 ($C_1$ Vertex sub-particle with TEC and λ1), EMD-31184 ($C_1$ Vertex sub-particle with μ1 and λ2), EMD-31187 (Asymmetric ISVP), and EMD-31188 (Icosahedral virion). The atomic coordinates have been deposited in the Protein Data Bank under accession codes 7ELH ($C_1$ Vertex sub-particle with TEC and λ1) and 7ELL ($C_1$ Vertex sub-particle with μ1 and λ2). Other data are available from the corresponding author upon reasonable request.

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

## Acknowledgements

We thank Yanxiang Cui for advice in both cryoEM and sub-particle reconstruction, Kang Zhou for assistance in cryoEM, and Titania Nguyen and Xiaoguo Liu for editorial assistance. This work was supported in part by grants from the National Key Research and Development Project of China (2016YFA0400900 to L.W.) and the US National Institutes of Health (AI094386 and GM071940 to Z.H.Z.). We acknowledge the use of instruments at the Electron Imaging Center for Nanomachines supported by UCLA and by instrumentation grants from NIH (1S10RR23057, 1S10OD018111 and U24GM116792), NSF (DBI-1338135 and DMR-1548924). As a visiting student at UCLA, M.P. was partially supported by a fellowship from China Scholarship Council (201804910804).

## Author contributions

Z.H.Z. and M.P. conceived the project; M.P. and A.L.A. prepared sample; A.L.A. made cryoEM grids; M.P. performed cryoEM imaging, determined the structures and built the atomic models; Z.H.Z., M.P., J.S.K., and A.L.A. interpreted the results and wrote the paper; Z.H.Z. supervised the research; L.W. and C.F. co-supervised M.P. while in China; all authors reviewed and approved the paper.

## Competing interests

The authors declare no competing interests.
