## [Peer Review File · Nature Communications]

REVIEWER COMMENTS

Reviewer #1 (Remarks to the Author):

Human reoviruses belong to a family of non-enveloped double-stranded RNA viruses that use membrane penetration proteins to perforate cell membranes and to deliver transcriptionally active viral nanoparticles into the cytoplasm of their host cells. Those viral nanoparticles not only contain the genomic RNA, but also enzymes for transcription and capping of newly synthesized viral mRNA. In the study by Pan et al. presented here, the authors applied state-of-the-art cryoEM methods, excellently executed, to reveal the architecture of infectious human reovirus particles and the structural organization of their transcription enzyme complexes (TECs) within the virion in so far not attained detail. They show how TECs are built from an RNA-dependent RNA polymerase (RdRp), the reovirus lambda3 protein, and a tightly-bound NTPase, the reovirus mu2 protein, for which we are able to see its atomic structure for the first time. The structures explain how flexible sequences from the capsid shell proteins (CSPs) can adopt multiple conformations to allow TEC integration inside the virus. We see, from an asymmetric reconstruction of the entire virion, how the double-stranded genomic RNA is organized and how it locally interacts with the TEC, which presumably is in an initiation-competent state. Additional insight into the assembly of the mu1 penetration protein layer and the incorporation of the sigma1 receptor binding protein are also presented. The work complements and expands on previous structural studies of related viruses (e.g. of cytoplasmic polyhedrosis virus [CPV], Cui et al., *Nat. Struct. Mol. Biol.*, 2019, reference 21; and of aquareovirus [ARV], Ding et al., *J. Virol.*, 2018, reference 22).

The research is presented well, with beautifully designed and self-explanatory figures depicting the structural observations. The interpretation of the results is appropriate and sufficient methodological details are provided (except how the RNA transcript strand is modeled and how the receptor-binding protein sigma1 trimer is modeled and interpreted; issues that can be addressed by the authors). This study is an important step forward in our understanding the biology of human reoviruses. I do have a few comments.

Major comments and suggestions:

The introduction would benefit from a brief description of what the biological function of the mu2 NTPase is.

Can the authors give a structural explanation why the TECs are organized with D3 symmetry (apart from the unoccupied vertices) within the virion? I was already curious about this when reading the Ding et al., *J. Virol.*, 2018, aquareovirus paper (reference 22). In rotavirus, the polymerases appear to bind randomly at each vertex without correlation between neighboring vertices, but in reovirus there appears to be some sort of cross talk between TECs to establish the observed D3 symmetry. Can this be rationalized by the conformation of the N termini and flexible loops of the CSPs? And a related question would be how do the CSPs of the polar and tropical vertices differ? What determines that only tropical vertices are unoccupied?

Line 215, the authors suggest the “front-mid” RNA duplex (red in Fig. 4a) to be the tail portion of the same genomic RNA segment for which the other end forms the branch (transcription fork) at the RdRp. What is the evidence for this conclusion? Why this one and not one of the other modeled RNA duplexes? And how would one know that it is indeed a tail

and not just an internal RNA segment?

I agree with how the template RNA strand was modeled. The model is supported by the experimental density map and its conformation is essentially identical to the template strand as observed in the crystal structure of the RdRp initiation complex (PDB 1N1H). However, the model of the RNA transcript strand (cyan in Fig. 4j-l) is clearly wrong. This is evident from the following observations:

- The model does not fit the experimental density.
- The model has impossible stereochemical distortions in the phosphodiester bonds.
- There is no base pairing between template and transcript nucleotides.
- Corresponding nucleotides in the crystal structure of the initiation complex (PDB 1N1H) are in entirely different conformations.

The authors should explore whether their density of the transcript strand can be interpreted based on the crystal structure of the initiation complex. Or, if the map in this region is of insufficient quality (as it appears to me) to accurately model these transcript nucleotides (if they are actually there), they should consider removing the transcript strand from the TEC model.

An RdRp crystal structure in complex with a cap analog (PDB-ID 1MWH) suggested that there is a cap-binding site on the surface of the polymerase. Do the authors perhaps see density at the corresponding position that would be indicative of a bound cap from the non-template (+)RNA 5' end?

Modeling and interpretation of the receptor-binding protein sigma1 trimer and capping enzyme lambda2 pentamer (lines 200-313 and Fig. 7c,f,g): According to Supplementary Fig. 3a, the reconstruction showing the sigma1 trimer density was obtained after classification, without alignment, from the particle stack obtained after sub-particle extraction at 5-fold vertices followed by C5 symmetry expansion. This means that the two classes (with and without sigma1 density) are both icosahedrally averaged and do have 5-fold symmetry imposed. It also means that the sigma1 density is the average of 5 distinct conformations and fitting of the sigma1 trimer into this 5-fold averaged density is not warranted (unless the superimposed conformations can be distinguished, which I doubt from the density shown in Fig. 7a). This leads to the following questions:

- 1) Have the authors attempted to obtain a local reconstruction of the sigma1 trimer (same protocol as for the TEC)?
- 2) There are two options for how binding of the sigma1 trimer on the lambda2 pentamer is guided: (i) any of the 5 possible positions is equally likely and there is no correlation between vertices and/or TEC binding. (ii) Sigma1 trimer binding is correlated to binding of the TEC complex at each vertex.
- 3) How does the sigma1 density appear in the asymmetric (C1) reconstruction of the ISVP, or in the sub-particle reconstruction of the TEC? Does it look 5-fold symmetric or asymmetric?
- 4) Does binding of the TEC at the inside of the 5-fold vertex translate into asymmetry within the lambda2 pentamer where sigma1 binds?
- 5) A classification for +/- sigma1 should be done with the TEC-aligned sub-particle stack.

Is there an obvious path for an mRNA transcript from the RdRp active site to the capping enzyme in the asymmetric reconstruction?

Supplementary Figure 5h, MRV vs. CPV, how were the two structures superimposed, e.g. which residues? From the figure, one can barely identify common structural features for the

two structures. Can the authors confirm that the superposition is correct (the MRV vs. ARV superposition does look correct).

Minor comments:

Title, "other proteins" sounds unspecific, maybe the authors can come up a more specific title?

Line 2, capitalize Orthoreovirus (Google: "a principal taxonomic category that ranks above species and below family, and is denoted by a capitalized Latin name, e.g. Leo.")

Line 8, NTPase should be defined?

Line 12, I suggest to add that they are "organized with different pseudo-D3 symmetries within the virion".

Line 28, "Spinoreovirinae subfamily" instead of "Sedoreovirinae subfamily"

Line 32, format reference?

Line 37, a reference to the crystal structures of the reovirus polymerase could be added here (Tao et al., Cell, 2002, reference 18).

Line 45, of asymmetrically organized elements

Line 54, "novel" should be deleted.

Lines 76, 102, 128, 170, 181, 320, 430, 436, 438, format reference?

Line 133, bound

Line 187, versus?

Line 214, it would be helpful to better specify "terminal dsRNA", e.g. 3' end of the (-)RNA template strand and the 5' end of the (+)RNA non-template strand.

Lines 232, 243, 265, please delete "Newly"

Line 266, which crystal structures?

Line 269, asymmetric unit of the icosahedral assembly

Line 467, I would change it to "we subtracted density from the particle images".

Line 472, "center of the"

Line 506, "symmetry-related" instead of "redundant"

Line 538, the number 254,390 differs from 102,966 shown in Supplementary Figure 3. 102,966 sounds about correct, as the theoretically expected number would be $14115 \cdot (60/5) \cdot (10/12) = 141150$, assuming polymerase occupancy at 10 out of the 12 five-

fold vertices.

Line 544, the formulation of the I3 icosahedral symmetry setting is confusing here. One shouldn't say that a plane goes through an axis. According to the Relion wiki: "5-fold axis on Z and 2-fold on Y. With the positive Z pointing at the viewer and without taken into account the 5-fold vertex in Z, there is one of the front-most 5-fold vertices in -XZ plane". It would also be helpful, if the direction of the X, Y, and Z axis were added to Supplementary Figure 3.

Line 573, threefold

Line 576, "By adjusting the volume threshold of core reconstruction", which core reconstruction, the one from the previous step with 35,354 core particles and C1 symmetry? Please specify.

Caption, Fig. 1: ISVP asymmetric core?

Reviewer #2 (Remarks to the Author):

Mammalian Reovirus (MRV), a mammal pathogen that infects gastrointestinal and respiratory tracks, is the prototypic member of the orthoreovirus genus. Due to its ability to infect humans is probably, together with Rotavirus and Bluetongue Virus, one of the more relevant members of the Reoviridae family. Despite the fact that MRV was among the first viral systems analyzed by cryoEM and the amount of structural information about its individual proteins, a complete high-resolution structure of the viral particle is still lacking. In this manuscript, Pan et al. resolve the asymmetric structure of MRV structure of the infectious subvirion particle at 3.3 Å resolution including the capsid proteins, the RNA-dependent RNA polymerase (RdRp) $\lambda 3$ and the NTPase $\mu 2$. Additionally to resolve, for the first time in my knowledge, the atomic structure of the RdRp cofactor $\mu 2$, the show that it is an NTPase and characterize the in situ structure and interactions viral proteins, genomic dsRNA and messenger ssRNA. The authors are world class leaders in the field of cryoEM and dsRNA virus structure. Specifically, they have been pioneers in the in situ resolution of the transcription machinery of different dsRNA virus. This work, not only provides relevant information for the dsRNA virus field but also provides a new detailed workflow for the resolution of similar complexes in other systems. The interest and novelty of the results support the publication of the manuscript, which could be published in its present form.

Minor comments:

- Figure 1. Indicate the correspondence between pentamers in Figure 1b and 1c (i.e. with numbers in 1c) would help the reader to follow the figure.
- p6 l189. Mercator projection is not a regular representation in the structural biology field. A explanation slightly more extensive than "earth-like" would help the reader to understand what is exactly shown. It must be indicated which points of the 3DR corresponds to the equator and poles in the representation.
- p6 l102 and pp128 l128. Ref 22 is bad formatted.

Reviewer #3 (Remarks to the Author):

$\mu 2$ has been a target of high interest for quite some time; Pan *et al.* present not only the authentic in-virion structure of this protein but also new insights into the receptor-binding factors, the interactions between the compacted genome and the RdRp, and the asymmetric architecture of the overall virion. While the Zhou lab's previous work on CPV established the methods used herein and foreshadowed some of the conclusions, mammalian orthoreovirus far and away the most important virus in this subfamily due in no small part to its ability to infect humans. I have strong enthusiasm about the experiments reported herein and many of the conclusions. However, there are three areas in which the authors have reached beyond the limitations of their data to make conclusions I feel are unsupported. The remaining conclusions are sufficiently exciting on their own to be of significant interest to the field.

Major comments:

1. The conclusion from the fully-asymmetric map that MRV has a different RdRp arrangement than CPV is suspect. Because the symmetry expansion is iterative and relies on first breaking local symmetry at one site, then locating the three-fold pseudoaxis, then expanding along that axis, particle misalignments in the final asymmetric structure are non-random. There is some low-occupancy density for the RdRp at both "empty" sites and this low-occupancy ghost density is not fivefold-symmetric. This is not inherently problematic as one would expect something like that when a small subset of particles fail to align, but it does little to bolster my confidence that the MRV definitely differs from the CPV arrangement.

If you apply D_3 symmetry to CPV and to the proposed MRV structure you will get the same arrangement. As Z. Hong Zhou has previously written in the pioneering work on asymmetric reconstruction of CPV, it is easy to get stuck in a local minimum with pseudosymmetry-breaking. This raises the question: is it more likely that MRV differs from other 10-pol reos in its configuration, or that the pseudosymmetry-breaking was biased? Based on the methods reported in Supp. Fig. 3 and in the text, I favor the latter explanation. Because the local arrangement of nearest-neighbors is non-random and is used to "bootstrap" to a fully asymmetric configuration, it would bias towards some asymmetric architectures and away from others.

It is conceivable that an analysis of the `rlnMaxValueProbDistribution` from the star files of the first symmetry-expansion phase could provide support or contradiction, or that an unbiased symmetry-expansion could be performed, or that another method could be used. Or, the authors could simply remove the overclaimed conclusions and the paper would still be quite good.

I note that this objection in no wise undermines the correctness of any of the atomic models, which would not be affected by this bias.

2. Based on existing figures, although the dominant mode of $\sigma 1$ interaction is clear, I strongly suspect that the resolution of $\sigma 1$ is insufficient to make these claims about R48

specifically. As a caveat, I have not reviewed the map data so it is *possible* this is merely a deficiency of the visuals and the authors could substantiate their claims with, *e.g.*, a supplementary movie.

3. The story about how these viruses invented different TEC arrangements when they independently developed genes is extremely speculative. See my further comments on l. 327-335.

Minor comments:

Why is >90% of the virus ISVP? Is this typical? When you say the pellet consisted mainly of virion particles (l. 413), do you mean authentic virion or do you mean ISVP? In l. 367, can you conclude that conversion to ISVP happens during purification, or could it be at or near release? l. 2 cell-free transcription also has been achieved in non-reos (for example, Molla 1991) so please rephrase this slightly

Introduction: Too much ink is spilt on shortcomings of previous studies; could trim that.

line 32 and elsewhere: some of your references did not convert to footnotes

l. 37 cite Tao 2002 (ref 18)

l. 101-103 & l. 317 It is not correct that the cited viruses are the only ones whose TEC organization is determined to date. The organization was also reported for a 9-segmented member of this subfamily (Kaelber 2020), which claimed to be an exception to the trend noted in l. 110-112

l. 208 Any potential role for ions? Is the spacing consistent or inconsistent with bridging?

l. 229 at transcription initiation state -> at a transcription initiation state

l. 230 one nucleotides -> one nucleotide

l. 253-254 Have the authors considered whether the N-terminal fragments could be involved in RdRp activation, as in rotavirus (McDonald & Patton, 2010; & earlier publications by the Patton lab)?

l. 327 Absence of evidence for homology may not be evidence of absence of homology due to the extreme divergence. After all, even homology in the highly-conserved CSP is not evident at the level of sequence alone when comparing distant family member (*e.g.*, rotavirus vs. orthoreovirus). If the authors wish to push this claim, they should present some bioinformatic evidence at least.

l. 333 This is a logical fallacy. The *de novo* emergence or horizontal transfer of a gene does not imply that the segment itself emerged independently as there exist several plausible mechanisms whereby the UTR origin would be conserved, including but not limited to recombination, duplication and hypermutation, and overprinting.

l. 334-335 Unclear what is meant by “independent incorporation of TEC”; see previous comment however.

l. 371 member -> membrane

l. 385 advices -> advice

l. 440 0.2 as a fraction of Nyquist or nm⁻¹?

I. 464-468 Can you clarify if classification without subtraction, using the exact same parameters otherwise, was attempted and failed, or are you going on theoretical considerations?

I. 574 fragment

Fig. 2B hard to tell dashed from dotted line. Change color or some other parameter for one of them?

Fig. 4I & S7 No-template -> non-template

Gamma phosphate density and densities in the surrounding residues is hard to see in the small 2D inset. Could this be overlaid in Supp. Fig. 5D or elsewhere?

The supplementary movies are quite nice.

As a typesetting comment, please use Schönflies notation for your point groups, e.g. “subjected to C_5 symmetry expansion” instead of “subjected to C5 symmetry expansion”. Relion input parameters are correct as-is.

Responses to Reviewers (NCOMMS-21-04549 by Pan *et al.*)

We would like to express our gratitude to all three reviewers for the quick and insightful critiques, and for supporting our work. As you will see from our itemized responses below, we have fully addressed the reviewers' questions and improved the clarity of the figures. The manuscript has been revised accordingly with major changes in the third paragraph of **Introduction** and first paragraph of **Discussion** (in response to Reviewer #3). In response to Reviewer #3's technical question, we have improved the clarity of our Supplementary Figure 3 and carried out more data processing that addressed Reviewer #3's concern regarding bias. To facilitate your navigation, the totality of the original reviews is copied in **black**, and our responses are shown in **blue**. The line numbers of changed text in the revised manuscript are indicated at the end of each answer (Ans). For your convenience of comparison, we also included a PDF file with text changes highlighted in **red** and deletions in **purple** with strikethrough.

Reviewer #1 (Remarks to the Author):

Human reoviruses belong to a family of non-enveloped double-stranded RNA viruses that use membrane penetration proteins to perforate cell membranes and to deliver transcriptionally active viral nanoparticles into the cytoplasm of their host cells. Those viral nanoparticles not only contain the genomic RNA, but also enzymes for transcription and capping of newly synthesized viral mRNA. In the study by Pan *et al.* presented here, the authors applied state-of-the-art cryoEM methods, excellently executed, to reveal the architecture of infectious human reovirus particles and the structural organization of their transcription enzyme complexes (TECs) within the virion in so far not attained detail. They show how TECs are built from an RNA-dependent RNA polymerase (RdRp), the reovirus lambda3 protein, and a tightly-bound NTPase, the reovirus mu2 protein, for which we are able to see its atomic structure for the first time. The structures explain how flexible sequences from the capsid shell proteins (CSPs) can adopt multiple conformations to allow TEC integration inside the virus. We see, from an asymmetric reconstruction of the entire virion, how the double-stranded genomic RNA is organized and how it locally interacts with the TEC, which presumably is in an initiation-competent state. Additional insight into the assembly of the mu1 penetration protein layer and the incorporation of the sigma1 receptor binding protein are also presented. The work complements and expands on previous structural studies of related viruses (e.g. of cytoplasmic polyhedrosis virus [CPV], Cui *et al.*, *Nat. Struct. Mol. Biol.*, 2019, reference 21; and of aquareovirus [ARV], Ding *et al.*, *J. Virol.*, 2018, reference 22).

The research is presented well, with beautifully designed and self-explanatory figures depicting the structural observations. The interpretation of the results is appropriate and sufficient methodological details are provided (except how the RNA transcript strand is modeled and how the receptor-binding protein sigma1 trimer is modeled and interpreted; issues that can be addressed by the authors). This study is an important step forward in our understanding the biology of human reoviruses. I do have a few comments.

Ans: Thank you for your kind words. Your comments are greatly appreciated and fully addressed below.

Major comments and suggestions:

1. The introduction would benefit from a brief description of what the biological function of the mu2 NTPase is.

Ans: We added one sentence, see lines 57-60.

2. Can the authors give a structural explanation why the TECs are organized with D_3 symmetry (apart from the unoccupied vertices) within the virion? I was already curious about this when reading the Ding *et al.*, *J. Virol.*, 2018, aquareovirus paper (reference 22). In rotavirus, the polymerases appear to bind randomly at each vertex without correlation between neighboring vertices, but in reovirus there appears to be some sort of cross talk between TECs to establish the observed D_3 symmetry. Can this be rationalized by the conformation of the N termini and flexible loops of the CSPs? And a related question would be how do the CSPs of the polar and tropical vertices differ? What determines that only tropical vertices are unoccupied?

Ans: We don't know why, but it does not have to be D_3 for other viruses in theory. There are two variations of D_3 — D_{3d} and D_{3h} , the latter was not possible because mirror symmetry does not work in biology. If we may offer

a philosophic speculation, D_{3d} is one of the possibilities and these viruses happened to encounter D_{3d} as a solution and they did not “bother” to change because it worked (the gist of Darwinism). Along the same vein, other members might use a different arrangement [indeed, applying the same strategy that led to a D_3 structure of CPV^[1] did not lead to an interpretable D_3 reconstruction for rotavirus^[2] and BTV^[3]. In fact, for the current work, applying the previously successful approach in data processing did not work for MRV, likely due to MRV’s multiple capsid layers that diminishes the contribution from TEC and RNA. Nonetheless, using a tedious approach of aligning TECs vertex by vertex (revised Supplementary Fig. 3a, b), we ended up with the D_3 organization found in CPV.

3. Line 215, the authors suggest the “front-mid” RNA duplex (red in Fig. 4a) to be the tail portion of the same genomic RNA segment for which the other end forms the branch (transcription fork) at the RdRp. What is the evidence for this conclusion? Why this one and not one of the other modeled RNA duplexes? And how would one know that it is indeed a tail and not just an internal RNA segment?

Ans: Two RNA duplexes were observed—one (*i.e.*, the cap-terminal RNA) near the template entrance and the other in the front-mid positions—in CPV previously^[4] and in MRV in the current study, at analogous positions around their TEC. The reason for assigning the “front-mid” density as we did in the current paper was explained exhaustively in the previous work^[4] (**Rebuttal figure 1**). According to the ouroboros model for transcription in dsRNA viruses (**Rebuttal figure 1**), RNA segments undergo continuous transcription with the cap-terminus RNA following the tail-terminal RNA (*i.e.*, during *elongation*, the re-annealed cap-terminal RNA chases the tail-terminal RNA). This ouroboros model is supported by the reconstructions of CPV RdRp in multiple states^[4]. As shown in **Rebuttal figure 1** below, the assignment of the orange-colored RNA (*i.e.*, “front-mid” RNA) as the tail-terminal RNA is based on two observations: first, it terminates near the TEC; second, it moved at *initiation* and disappeared at *elongation* states. In our MRV structure, the “front-mid” density also terminates near TEC. According to the positional conservation of the “front-mid” RNA duplex, we assigned it as the tail-terminal of the same genomic RNA segment, the other end of which forms the branch (*i.e.*, the transcription fork).

4. I agree with how the template RNA strand was modeled. The model is supported by the experimental density map and its conformation is essentially identical to the template strand as observed in the crystal structure of the RdRp initiation complex (PDB 1N1H). However, the model of the RNA transcript strand (cyan in Fig. 4j-l) is clearly wrong. This is evident from the following observations:

- The model does not fit the experimental density.
- The model has impossible stereochemical distortions in the phosphodiester bonds.
- There is no base pairing between template and transcript nucleotides.
- Corresponding nucleotides in the crystal structure of the initiation complex (PDB 1N1H) are in entirely different conformations.

The authors should explore whether their density of the transcript strand can be interpreted based on the crystal structure of the initiation complex. Or, if the map in this region is of insufficient quality (as it appears to me) to accurately model these transcript nucleotides (if they are actually there), they should consider removing the transcript strand from the TEC model.

Ans: We agree with your suggestion. As such, we have removed the transcript atomic model and named the corresponding density as putative transcript. See lines 254-256, 720-724, and 834.

5. An RdRp crystal structure in complex with a cap analog (PDB-ID 1MWH) suggested that there is a cap-binding site on the surface of the polymerase. Do the authors perhaps see density at the corresponding position that would be indicative of a bound cap from the non-template (+)RNA 5' end?

Ans: We didn't see a density at the corresponding position. For our density, the (+)RNA 5' end is at the upper edge of the template exit channel (revised Fig. 4i and Supplementary Fig. 7d), rather than between the N-terminal domain and the thumb domain, suggesting that the current structure is at a stage later than the capping-binding stage (as a support of this interpretation, capping binding has been visualized in CPV at transcription initiation state, see **Rebuttal figure 1** above and Cui et al. 2019).

6. Modeling and interpretation of the receptor-binding protein sigma1 trimer and capping enzyme lambda2 pentamer (lines 200-313 and Fig. 7c,f,g): According to Supplementary Fig. 3a, the reconstruction showing the sigma1 trimer density was obtained after classification, without alignment, from the particle stack obtained after sub-particle extraction at 5-fold vertices followed by C5 symmetry expansion. This means that the two classes (with and without sigma1 density) are both icosahedrally averaged and do have 5-fold symmetry imposed. It also means that the sigma1 density is the average of 5 distinct conformations and fitting of the sigma1 trimer into this 5-fold averaged density is not warranted (unless the superimposed conformations can be distinguished, which I doubt from the density shown in Fig. 7a). This leads to the following questions:

1) Have the authors attempted to obtain a local reconstruction of the sigma1 trimer (same protocol as for the TEC)?

Ans: Yes, we obtained the sub-particle reconstruction of σ_1 trimer by following the same protocol as for the TEC. Briefly, we classified the sub-particles of σ_1 and λ_2 into six reconstruction classes. One (40.9% sub-particles) of six reconstructions didn't show the density of σ_1 , the remaining five reconstructions all showed densities corresponding to σ_1 and these densities are C_5 related. The proportions of the five classes with σ_1 densities are 10.7%, 13.7%, 11.1%, 11.4%, 12.2%. These reconstructions are shown in the revised Supplementary Fig. 3.

2) There are two options for how binding of the sigma1 trimer on the lambda2 pentamer is guided: (i) any of the 5 possible positions is equally likely and there is no correlation between vertices and/or TEC binding. (ii) Sigma1 trimer binding is correlated to binding of the TEC complex at each vertex.

Ans: We used the first option, that is, any of the 5 possible positions is equally likely and there is no correlation between vertices and TEC binding. The evidence is shown below (see answers to sub-questions 3 through 5).

3) How does the sigma1 density appear in the asymmetric (C1) reconstruction of the ISVP, or in the sub-particle reconstruction of the TEC? Does it look 5-fold symmetric or asymmetric?

Ans: The σ_1 density appears in the high thresholds of both the asymmetric reconstruction of ISVP and the sub-particle reconstruction of TEC (**Rebuttal figure 2**). In the asymmetric reconstruction of ISVP, the resolution of the σ_1 density is insufficient to tell the symmetry due to binning. In the C_1 TEC sub-particle reconstruction, the σ_1 density appears 5-fold symmetric.

Rebuttal figure 2. Surface representation of the σ_1 density in asymmetric ISVP reconstruction (left) and TEC-aligned sub-particle reconstruction (right).

4) Does binding of the TEC at the inside of the 5-fold vertex translate into asymmetry within the lambda2 pentamer where sigma1 binds?

Ans: No, it doesn't. There is no correlation between the TEC and $\sigma 1$ binding.

5) A classification for +/- sigma1 should be done with the TEC-aligned sub-particle stack.

Ans: We classified the TEC-aligned sub-particle stack focusing on the $\sigma 1$ density (**Rebuttal figure 3**). The result has no significant difference from our original result.

Is there an obvious path for an mRNA transcript from the RdRp active site to the capping enzyme in the asymmetric reconstruction?

Ans: Yes, there is a positively-charged tunnel from RdRp active site leading to the center of $\lambda 1A$ pentamer/capping enzyme turret at each TEC-occupied vertex (revised Supplementary Fig. 7e).

Supplementary Figure 5h, MRV vs. CPV, how were the two structures superimposed, e.g. which residues? From the figure, one can barely identify common structural features for the two structures. Can the authors confirm that the superposition is correct (the MRV vs. ARV superposition does look correct).

Ans: We superimposed the NTPase proteins of MRV and CPV by matchmaker command in UCSF Chimera X (mm #ARV_NTPase_model to #MRV_NTPase_model). As a result, the conserved NTPase domains of MRV and CPV are well superimposed (we have inserted an overview of the superimposed structure to make this point clear in the revised Supplementary Fig. 5h).

Minor comments:

Title, "other proteins" sounds unspecific, maybe the authors can come up a more specific title?

Ans: Fixed. We modified title to "Asymmetric reconstruction of mammalian reovirus reveals interactions among RNA, transcriptional factor $\mu 2$ and capsid proteins".

Line 2, capitalize Orthoreovirus (Google: "a principal taxonomic category that ranks above species and below family, and is denoted by a capitalized Latin name, e.g. Leo.")

Ans: Fixed. We capitalized "Orthoreovirus" in line 2, line 20, and line 78.

Line 8, NTPase should be defined?

Ans: Fixed. We now spell out "NTPase" as "nucleoside triphosphatase" in line 8.

Line 12, I suggest to add that they are "organized with different pseudo-D3 symmetries within the virion".

Ans: The location of RNA polymerase complexes inside virion is implied. We took advantage of this to avoid explicitly stating "within the virion" so that we could keep the summary within the 150 words limit dictated by the journal. If the editor would make an exception to this limit, we would gladly add them.

Line 28, "Spinoreovirinae subfamily" instead of "Sedoreovirinae subfamily"

Ans: Fixed. Thank you for this great catch! See line 29.

Line 32, format reference?

Ans: Good catch, but this is following the journal requirement to prevent confusion of a superscript reference number on a number [e.g., $\sigma 3^{[5]}$ vs $\sigma 3$ (Ref 22)].

Line 37, a reference to the crystal structures of the reovirus polymerase could be added here (Tao et al., Cell, 2002, reference 18).

Ans: Fixed. This reference is added now. See line 38 (ref 16).

Line 45, of asymmetrically organized elements

Ans: Fixed. See line 57.

Line 54, "novel" should be deleted.

Ans: Done. See line 73.

Lines 76, 102, 128, 170, 181, 320, 430, 436, 438, format reference?

Ans: Good catch, but this is following the journal requirement to prevent confusion of a superscript reference number on a number [e.g., $\sigma 3^{[5]}$ vs $\sigma 3$ (Ref 22)].

Line 133, bound

Ans: Fixed. See line 163.

Line 187, versus?

Ans: Fixed. See lines 217-218.

Line 214, it would be helpful to better specify “terminal dsRNA”, e.g. 3’ end of the (-)RNA template strand and the 5’ end of the (+)RNA non-template strand.

Ans: Fixed. See lines 247-248.

Lines 232, 243, 265, please delete “Newly”

Ans: Fixed. See lines 264, 275, 302.

Line 266, which crystal structures?

Ans: Fixed. See in reference 13 and 14.

Line 269, asymmetric unit of the icosahedral assembly

Ans: Fixed. See lines 306-307.

Line 467, I would change it to “we subtracted density from the particle images”.

Ans: Fixed. See line 510.

Line 472, “center of the”

Ans: Fixed. See line 516.

Line 506, “symmetry-related” instead of “redundant”

Ans: Fixed. See line 549.

Line 538, the number 254,390 differs from 102,966 shown in Supplementary Figure 3. 102,966 sounds about correct, as the theoretically expected number would be $14115 \cdot (60/5) \cdot (10/12) = 141150$, assuming polymerase occupancy at 10 out of the 12 five-fold vertices.

Ans: Fixed. Thank you for the great catch! The number 254,390 is from an old set of data, and we had forgotten to change it in our original submission.

Line 544, the formulation of the I3 icosahedral symmetry setting is confusing here. One shouldn’t say that a plane goes through an axis. According to the Relion wiki: “5-fold axis on Z and 2-fold on Y. With the positive Z pointing at the viewer and without taken into account the 5-fold vertex in Z, there is one of the front-most 5-fold vertices in -XZ plane”. It would also be helpful, if the direction of the X, Y, and Z axis were added to Supplementary Figure 3.

Ans: Fixed. See lines 587 and 591 in revised manuscript and the revised Supplementary Fig. 3a. (The 5-fold vertex in Z is Vertex 1, and the vertex in -XZ plane is Vertex 2.)

Line 573, threefold

Ans: Fixed. See line 616.

Line 576, “By adjusting the volume threshold of core reconstruction”, which core reconstruction, the one from the previous step with 35,354 core particles and C1 symmetry? Please specify.

Ans: Fixed. See line 614. Yes, the core reconstruction is refined from 35,354 core particles with C_1 symmetry.

Caption, Fig. 1: ISVP asymmetric core?

Ans: Fixed. We changed these words to “RNA genome and TEC”.

Thank you for being so careful and for catching these typos, which, as detailed above, have all been fixed in the revised manuscript.

Reviewer #2 (Remarks to the Author):

Mammalian Reovirus (MRV), a mammal pathogen that infects gastrointestinal and respiratory tracks, is the prototypic member of the orthoreovirus genus. Due to its ability to infect humans is probably, together with Rotavirus and Bluetongue Virus, one of the more relevant members of the Reoviridae family. Despite the fact that MRV was among the first viral systems analyzed by cryoEM and the amount of structural information about its individual proteins, a complete high-resolution structure of the viral particle is still lacking. In this manuscript, Pan et al. resolve the asymmetric structure of MRV structure of the infectious subviriion particle at 3.3 Å resolution including the capsid proteins, the RNA-dependent RNA polymerase (RdRp) $\lambda 3$ and the

NTPase $\mu 2$. Additionally to resolve, for the first time in my knowledge, the atomic structure of the RdRp cofactor $\mu 2$, the show that it is an NTPase and characterize the in situ structure and interactions viral proteins, genomic dsRNA and messenger ssRNA. The authors are world class leaders in the field of cryoEM and dsRNA virus structure. Specifically, they have been pioneers in the in situ resolution of the transcription machinery of different dsRNA virus. This work, not only provides relevant information for the dsRNA virus field but also provides a new detailed workflow for the resolution of similar complexes in other systems. The interest and novelty of the results support the publication of the manuscript, which could be published in its present form.

Ans: Thank you for your generous support!

Minor comments:

- Figure 1. Indicate the correspondence between pentamers in Figure 1b and 1c (i.e. with numbers in 1c) would help the reader to follow the figure.

Ans: Fixed. See the revised Fig. 1b.

- p6 ll89. Mercator projection is not a regular representation in the structural biology field. A explanation slightly more extensive than “earth-like” would help the reader to understand what is exactly shown. It must be indicated which points of the 3DR corresponds to the equator and poles in the representation.

Ans: Fixed. See lines 110-112.

- p6 ll102 and pp128 ll128. Ref 22 is bad formatted.

Ans: Good catch, but this is following the journal requirement to prevent confusion of a superscript reference number on a number [e.g., $\sigma 3^{[5]}$ vs $\sigma 3$ (Ref 22)].

Reviewer #3 (Remarks to the Author):

$\mu 2$ has been a target of high interest for quite some time; Pan et al. present not only the authentic in-virion structure of this protein but also new insights into the receptor-binding factors, the interactions between the compacted genome and the RdRp, and the asymmetric architecture of the overall virion. While the Zhou lab's previous work on CPV established the methods used herein and foreshadowed some of the conclusions, mammalian orthoreovirus far and away the most important virus in this subfamily due in no small part to its ability to infect humans. I have strong enthusiasm about the experiments reported herein and many of the conclusions. However, there are three areas in which the authors have reached beyond the limitations of their data to make conclusions I feel are unsupported. The remaining conclusions are sufficiently exciting on their own to be of significant interest to the field.

Ans: We greatly appreciate your generous support of our work. As you can see from our detailed responses below, the three areas you have identified all touch upon important challenges facing the cryoEM field and our efforts only represent some of the first attempts to tackle these challenging problems. We hope that you would agree that our results represent informative steps in the right direction, though by no means final answers to these problems.

Major comments:

1. The conclusion from the fully-asymmetric map that MRV has a different RdRp arrangement than CPV is suspect. Because the symmetry expansion is iterative and relies on first breaking local symmetry at one site, then locating the three-fold pseudoaxis, then expanding along that axis, particle misalignments in the final asymmetric structure are non-random. There is some low-occupancy density for the RdRp at both “empty” sites and this low-occupancy ghost density is not fivefold-symmetric. This is not inherently problematic as one would expect something like that when a small subset of particles fail to align, but it does little to bolster my confidence that the MRV definitely differs from the CPV arrangement.

If you apply D3 symmetry to CPV and to the proposed MRV structure you will get the same arrangement. As Z. Hong Zhou has previously written in the pioneering work on asymmetric reconstruction of CPV, it is easy to get stuck in a local minimum with pseudosymmetry-breaking. This raises the question: is it more likely that MRV differs from other 10-pol reos in its configuration, or that the pseudosymmetry-breaking was biased? Based on the methods reported in Supp. Fig. 3 and in the text, I favor the latter explanation. Because the local

arrangement of nearest-neighbors is non-random and is used to “bootstrap” to a fully asymmetric configuration, it would bias towards some asymmetric architectures and away from others.

It is conceivable that an analysis of the *rlnMaxValueProbDistribution* from the star files of the first symmetry-expansion phase could provide support or contradiction, or that an unbiased symmetry-expansion could be performed, or that another method could be used. Or, the authors could simply remove the overclaimed conclusions and the paper would still be quite good.

I note that this objection in no wise undermines the correctness of any of the atomic models, which would not be affected by this bias.

Ans: There seems to be some misunderstanding of our workflow as depicted in our original Supplementary Fig. 3. This is because the only part of that figure dealing with D_3 to “a fully asymmetric configuration” is the right lower corner portion of the workflow (part of revised Supplementary Fig. 3e, which is about 1/6 of the entire figure). All the other sections of the workflow illustrate what we had to do in order to successfully classify sub-particle (note, not full particle) reconstructions for a D_3 symmetric structure of the virus, starting from an icosahedral reconstruction through two stages of sub-particle classification (V1 and V2). We reckon that the confusion might be due to our labeling of the figures, in which panel b in the original figure contains both steps used for V2 sub-particle reconstruction in order to obtain a D_3 (symmetric) reconstruction and decoupling from D_3 to C_1 (asymmetric) structures for full-particle asymmetric reconstruction. We now have organized Supplementary Fig. 3 into five main sections (a through e) with different color shades to make icosahedral symmetric, sub-particle C_1 classifications (V1 and V2), decoupling from D_3 symmetric to full-particle C_1 asymmetric reconstruction steps unmistakably clear.

As you can see from the revised Supplementary Fig. 3, there is NO bootstrap step used from D_3 to C_1 . Instead, there is a straightforward D_3 to C_1 symmetry expansion step, followed by the full-particle C_1 3D classification step. The result is that, even when we requested 15 3D

classes for the full particle, only *one* structure emerged [that is, 6 structures (classes 1, 2, 9, 10, 12, 13 in the following table) among the 15 requested structures are identical and the remainders (classes 3, 4, 5, 6, 7, 8, 11, 14, 15 in the following table) are all junk]. The following table lists values of *rlnMaxValueProbDistribution* from the data STAR file of the last 3D classification step, which also indicates absence of bias among these nearly identical structures of the ISVP full particle reconstructions.

TEC and genome			Junk		
Class number	Average rlnMaxValueProbDistribution	Proportion (%)	Class number	Average rlnMaxValueProbDistribution	Proportion (%)
1	0.999919	11.1	3	0.999435	2.8
2	0.999883	10.7	4	0.999615	3.4
9	0.999890	11.3	5	0.999439	2.7
10	0.999877	11.7	6	0.999424	2.8
12	0.999877	10.6	7	0.999556	2.6
13	0.999873	11.5	8	0.999624	2.5
			11	0.999504	2.6
			14	0.999508	2.5
			15	0.999762	11.2

To test whether choosing only one V1 sub-particle class (revised Supplementary Fig. 3b) would lead to bias, we pooled all sub-particles together by rotating the pooled V1 sub-particle classes with RdRp (red boxes in revised Supplementary Fig. 3b) to the same orientation, and repeated both the V2 sub-particle classification step (revised Supplementary Fig. 3d) and the subsequent D3 symmetry decoupling step (revised Supplementary Fig. 3e). The result was the same as what we obtained before and revealed dsRNA densities at the two vertices without TEC (vertices 5 and 8 in **Rebuttal figure 4** above), suggesting choosing one of the redundant sub-particle classes for decoupling (at the end of revised Supplementary Fig. 3b) did not bias outcome.

2. Based on existing figures, although the dominant mode of $\sigma 1$ interaction is clear, I strongly suspect that the resolution of $\sigma 1$ is insufficient to make these claims about R48 specifically. As a caveat, I have not reviewed the map data so it is possible this is merely a deficiency of the visuals and the authors could substantiate their claims with, e.g., a supplementary movie.

Ans: Indeed, the resolution of this density is insufficient to ascertain accurate positioning of side chains, and the fitting of the $\sigma 1$ trimer atomic model into a map only represents a provisional model. As such, we have toned down all the statements describing the interactions in this paragraph, See lines 339-350. Our suggestion that the R48 residues of $\sigma 1$ interacting with D1283 residues of $\lambda 2$ is based on geometry and chemistry (Supplementary Movie 5). In our density, the end portion of $\sigma 1$ inside the capsid is clear and we used that to fit the $\sigma 1$ crystal structure. In the $\sigma 1$ crystal structure, three R48 residues are the only residues that point away from the three-fold axis and could form hydrogen bonds with the residues of $\lambda 2$.

3. The story about how these viruses invented different TEC arrangements when they independently developed genes is extremely speculative. See my further comments on l. 327-335.

Ans: Following your advice, we have removed the speculative statements. As detailed in our response to your comments on l. 369-375 below, we now acknowledge that the emergence of different TEC configurations needed more investigation in our revised manuscript.

Minor comments:

Why is >90% of the virus ISVP? Is this typical?

Ans: We are not sure, but suspect that this is caused by our purification method, maybe centrifugation at $100,000 \times g$ for 2 hr.

When you say the pellet consisted mainly of virion particles (l. 413), do you mean authentic virion or do you mean ISVP?

Ans: Fixed. Replace "virion particles" with "virus particles". See line 456.

In l. 367, can you conclude that conversion to ISVP happens during purification, or could it be at or near release?

Ans: Yes, we can conclude that conversion to ISVP happens during purification, because we collected all the ISVP outside of cells (from the supernatant of infected cells).

l. 27 cell-free transcription also has been achieved in non-reos (for example, Molla 1991) so please rephrase this slightly

Ans: Fixed. See line 27.

Introduction: Too much ink is spilt on shortcomings of previous studies; could trim that.

Ans: Fixed. Changed the tune to recognize the contribution of previous studies about predicting the locations of RdRp $\lambda 3$ and its cofactor $\mu 2$. See lines 38-56.

line 32 and elsewhere: some of your references did not convert to footnotes

Ans: Good catch, but this is following the journal requirement to prevent confusion of a superscript reference number on a number [e.g., $\sigma 3^{[5]}$ vs $\sigma 3$ (Ref 22)].

l. 37 cite Tao 2002 (ref 18)

Ans: This reference is added now. See line 38 (ref 16).

I. 101-103 & I. 317 It is not correct that the cited viruses are the only ones whose TEC organization is determined to date. The organization was also reported for a 9-segmented member of this subfamily (Kaelber 2020), which claimed to be an exception to the trend noted in I. 110-112

Ans: Good catch! Indeed, we missed this recent paper, which is now cited. See lines 126 and 130. Upon careful reading of the method presented in that paper, we see that the authors had to rely on the use of artificial models with different numbers and orientations of TECs (“decoys”) to cross-correlate with particles to find out probabilities of each kind of TEC configurations. Consequently, the authors could not have a unique answer as to which TEC configurations the virus has, but rather different (non-zero) probabilities of each configuration, with the 10 TEC configuration of the CPV kind having the highest probability (60%). We note that the decoy-based method the authors tried is different from our direct method used in our current paper and previous CPV papers. With our method, there is no indication or hint of other TEC configurations other than the one reported. The cryoEM image (Fig. 1a in the Kaelber et al. paper) shows a lot of particles devoid of RNA genome (“empty”) and lots of leaked out RNA duplexes in the background, suggesting many virion particles might be partially damaged, potentially releasing some or all RNA segments. This might have given rise to variable numbers of TEC inside virions and/or could obscure answers when their decoy-based method was used to calculate probabilities of different TEC configurations.

An important point of our discovery is that the capped end of the RNA segment is visualized directly inside the entrance of RNA transcription. We did not find any TEC *without* cap-terminal RNA attached to it, indicating that the numbers of segments and TEC is the same. In addition, it only makes biological sense that each TEC has its own specific RNA segment. Therefore, we believe there is still room to revisit this issue for the Fako virus, perhaps by processing cryoEM images of intact virions (other than “empty”, likely damaged particles) with better contrast than those used for the Kaelber *et al.* paper. Nonetheless, we thank you for bringing to our attention of this recent paper and we had now appropriately cited it in the revised paper.

I. 208 Any potential role for ions? Is the spacing consistent or inconsistent with bridging?

Ans: We don’t see potential density for ions. This might simply reflect that the resolution is not good enough to resolve ions.

I. 229 at transcription initiation state -> at a transcription initiation state

Ans: Fixed. See line 261.

I. 230 one nucleotides -> one nucleotide

Ans: Deleted. See line 263.

I. 253-254 Have the authors considered whether the N-terminal fragments could be involved in RdRp activation, as in rotavirus (McDonald & Patton, 2010; & earlier publications by the Patton lab)?

Ans: Thank you for pointing out this possibility! We now compared the N-terminal amphiphilic helix fragment that interacts with the RdRp and found that an RdRp interaction exists with the N-terminal amphiphilic helix fragment of $\lambda 1A_4$, similar to that shown to regulate transcription in rotavirus VP2. We added a sentence to suggest a possible similar regulatory role of this N-terminal fragment of $\lambda 1A_4$ as that shown for rotavirus VP2 and cited the corresponding papers. See lines 287-289.

I. 327 Absence of evidence for homology may not be evidence of absence of homology due to the extreme divergence. After all, even homology in the highly-conserved CSP is not evident at the level of sequence alone when comparing distant family member (e.g., rotavirus vs. orthoreovirus). If the authors wish to push this claim, they should present some bioinformatic evidence at least.

Ans: We fully agree with you that one can’t infer absence of structural homology from lack of recognized sequence homology. Fortunately, in the current case, we have atomic structures for all these proteins being compared. While the “conserved” proteins do share great structural similarities, as expected, the proteins we indicated to be non-conserved do not have any similarities in their atomic structures. We now make this point more clearly by emphasizing that our assertion is based on known atomic structures of these proteins, rather than protein sequences. See lines 366-368.

I. 333 This is a logical fallacy. The de novo emergence or horizontal transfer of a gene does not imply that the segment itself emerged independently as there exist several plausible mechanisms whereby the UTR origin would be conserved, including but not limited to recombination, duplication and hypermutation, and overprinting.

Ans: Thank you for this very insightful advice! We agree that there are too many possibilities and now acknowledge that the emergence of different TEC configurations need more investigation in our revised manuscript. See lines 373-379.

I. 334-335 Unclear what is meant by “independent incorporation of TEC”; see previous comment however.

Ans: Deleted. See line 375.

I. 371 member -> membrane

Ans: Fixed. See line 414.

I. 385 advices -> advice

Ans: Fixed. See line 428.

I. 440 0.2 as a fraction of Nyquist or nm^{-1} ?

Ans: nm^{-1}

I. 464-468 Can you clarify if classification without subtraction, using the exact same parameters otherwise, was attempted and failed, or are you going on theoretical considerations?

Ans: Yes, we can. We classified without subtraction and failed.

I. 574 fragment

Ans: Fixed. See lines 638-642.

Fig. 2B hard to tell dashed from dotted line. Change color or some other parameter for one of them?

Ans: Fixed. Made the vertical dotted lines thicker, and indicated the dashed lines are colored as domains.

Fig. 4I & S7 No-template -> non-template

Ans: Fixed.

Gamma phosphate density and densities in the surrounding residues is hard to see in the small 2D inset.

Could this be overlaid in Supp. Fig. 5D or elsewhere?

Ans: Fixed. The density was shown as mesh in revised Supplementary Fig. 5D.

The supplementary movies are quite nice.

Ans: Thank you!

As a typesetting comment, please use Schönflies notation for your point groups, e.g. “subjected to C_5 symmetry expansion” instead of “subjected to C5 symmetry expansion”. Relion input parameters are correct as-is.

Ans: We now have changed C5 to C_5 , C1 to C_1 , D3 to D_{3d} , but we leave I3 unchanged to comply with the Relion convention and avoid causing confusion to future investigators who might follow our workflow.

In **summary**, we are grateful to you for the careful reviews and the suggested improvements, which have been all incorporated in the revised manuscript and figures.

References:

- [1] X. Zhang, K. Ding, X. Yu, W. Chang, J. Sun, Z. H. Zhou, *Nature* **2015**, 527, 531-534.
- [2] K. Ding, C. C. Celma, X. Zhang, T. Chang, W. Shen, I. Atanasov, P. Roy, Z. H. Zhou, *Nat Commun* **2019**, 10, 2216.
- [3] Y. He, S. Shivakoti, K. Ding, Y. Cui, P. Roy, Z. H. Zhou, *Proceedings of the National Academy of Sciences* **2019**, 116, 16535-16540.
- [4] Y. Cui, Y. Zhang, K. Zhou, J. Sun, Z. H. Zhou, *Nat Struct Mol Biol* **2019**, 26, 1023-1034.
- [5] J. Jane-Valbuena, M. L. Nibert, S. M. Spencer, S. B. Walker, T. S. Baker, Y. Chen, V. E. Centonze, L. A. Schiff, *J Virol* **1999**, 73, 2963-2973.

REVIEWER COMMENTS

Reviewer #1 (Remarks to the Author):

The authors have carefully addressed the reviewers' comments and revised the manuscript accordingly. The new version of the manuscript is ready to be published.

Reviewer #3 (Remarks to the Author):

Except for Major Comment 1, the responses to all major comments address my concerns. The authors and I disagree on whether their protocol of “straightforward expansion from *D₃* to *C₁*” could introduce a bias. I argue that going through the *D₃* intermediate in this way can introduce bias. The fact that all 6 “good” classes from the 3D classification of symmetry-expanded particles had the same configuration does reduce the probability that this bias contributed to an erroneous result. Because the protocol is sufficiently well-described that an investigator could reproduce it, and because it seems unlikely we can come to a productive agreement via review rounds, I don't want to belabor overmuch the possibility that it is biased. To put it informally...the rebuttal does not fully convince me but I think it's more important for this work to be disseminated in a timely manner than to drill down on this point, as more work needs to be done in the subfield to fully clarify this issue.

Minor & orthography:

Italicize *Orthoreovirus* if you are capitalizing it. I remind reviewer #1 that it is acceptable practice in virology to use “orthoreovirus,” (without capitals) as a collective name rather than a taxon name, and therefore the original manuscript was technically not in error. The definitive guide to virus orthography is found at <https://talk.ictvonline.org/information/w/faq/386/how-to-write-virus-species-and-other-taxa-names>.

II. 110-112 does not actually address the request for clarification about Mercator. Be more specific about what was actually done to create the visual.

I. 483 you have stated nm⁻¹ in your rebuttal but the unit should be in the paper.

Final Responses to Reviewers (NCOMMS-21-04549 by Pan *et al.*)

We would like to express our gratitude to Reviewers 1 and 3 for the quick review of our revised paper. We provide a point-by-point responses below to their comments. To facilitate your navigation, the totality of the original reviews is copied in **black**, and our responses are shown in **blue**. The line numbers of changed text in the revised manuscript are indicated at the end of each answer (**Ans**).

Reviewer #1 (Remarks to the Author):

The authors have carefully addressed the reviewers' comments and revised the manuscript accordingly. The new version of the manuscript is ready to be published.

Ans: Thank you for your kind support!

Reviewer #3 (Remarks to the Author):

Except for Major Comment 1, the responses to all major comments address my concerns. The authors and I disagree on whether their protocol of “straightforward expansion from D_3 to C_1 ” could introduce a bias. I argue that going through the D_3 intermediate in this way can introduce bias. The fact that all 6 “good” classes from the 3D classification of symmetry-expanded particles had the same configuration does reduce the probability that this bias contributed to an erroneous result. Because the protocol is sufficiently well-described that an investigator could reproduce it, and because it seems unlikely we can come to a productive agreement via review rounds, I don't want to belabor overmuch the possibility that it is biased. To put it informally...the rebuttal does not fully convince me but I think it's more important for this work to be disseminated in a timely manner than to drill down on this point, as more work needs to be done in the subfield to fully clarify this issue.

Minor & orthography:

Ans: Thank you for support despite of your remaining concerning regarding possible bias! We are happy to acknowledge this difference in our opinions by adding the following two sentences near the end of the first paragraph of the Discussion section (Lines 371-376 in the final manuscript file):

“In addition, it remains controversial regarding possible bias in the various approaches used for decoupling asymmetric structures from the icosahedral arranged components. Indeed, the TEC organizations of Fako virus were not unique based on probabilities estimated by cross-correlating with various arrangements of TEC decoys²⁵ and those within non-turreted dsRNA viruses in the *Sedoreovirinae* subfamily of the *Reoviridae* are yet to be established.”

Italicize *Orthoreovirus* if you are capitalizing it. I remind reviewer #1 that it is acceptable practice in virology to use “orthoreovirus,” (without capitals) as a collective name rather than a taxon name, and therefore the original manuscript was technically not in error. The definitive guide to virus orthography is found at <https://talk.ictvonline.org/information/w/faq/386/how-to-write-virus-species-and-other-taxa-names>.

Ans: Thank you for the advice!

II. 110-112 does not actually address the request for clarification about Mercator. Be more specific about what was actually done to create the visual.

Ans: We now added an explanation of the Mercator projection (i.e., the surface of a sphere is projected into a rectangle map). Line 116.

I. 483 you have stated nm^{-1} in your rebuttal but the unit should be in the paper.

Ans: It should have been \AA^{-1} and is fixed in the manuscript now. Line 467

In **summary**, we are grateful to you for the careful reviews and the suggested improvements, which have been all incorporated in the final manuscript.